# MGDD: A Meta Generator for
# Fast Dataset Distillation

**Songhua Liu    Xinchao Wang**[*]
National University of Singapore
songhua.liu@u.nus.edu, xinchao@nus.edu.sg

## Abstract

Existing dataset distillation (DD) techniques typically rely on iterative strategies to synthesize condensed datasets, where datasets before and after distillation are forward and backward through neural networks a massive number of times. Despite the promising results achieved, the time efficiency of prior approaches is still far from satisfactory. Moreover, when different sizes of synthetic datasets are required, they have to repeat the iterative training procedures, which is highly cumbersome and lacks flexibility. In this paper, different from the time-consuming forward-backward passes, we introduce a generative fashion for dataset distillation with significantly improved efficiency. Specifically, synthetic samples are produced by a generator network conditioned on the initialization of DD, while synthetic labels are obtained by solving a least-squares problem in a feature space. Our theoretical analysis reveals that the errors of synthetic datasets solved in the original space and then processed by any conditional generators are upper-bounded. To find a satisfactory generator efficiently, we propose a meta-learning algorithm, where a meta generator is trained on a large dataset so that only a few steps are required to adapt to a target dataset. The meta generator is termed as *MGDD* in our approach. Once adapted, it can handle arbitrary sizes of synthetic datasets, even for those unseen during adaptation. Experiments demonstrate that the generator adapted with only a limited number of steps performs on par with those state-of-the-art DD methods and yields $22\times$ acceleration.

## 1 Introduction

Dataset distillation (DD) introduced by Wang *et al.* [51] aims to compress an original dataset $\mathcal{T}$ into a much smaller synthetic set $\mathcal{S}$, such that the performance of a neural network, trained with the condensed dataset $\mathcal{S}$, is similar to the network trained with $\mathcal{T}$. The derived synthetic datasets not only save the cost of storage and transmission but also significantly reduce the computational resources and time required by training models using original datasets. As such, DD finds its application across a wide spectrum of domains and is receiving increasing attention from the community.

The typical paradigm of DD is to optimize $\mathcal{S}$ in an iterative loop, as shown in Fig. 1(a). In each iteration, a new network is sampled, leveraged by which a matching metric is calculated for $\mathcal{S}$ and $\mathcal{T}$, and the matching loss is then back-propagated through the network to update $\mathcal{S}$. Recently, a large number of approaches have been dedicated to exploring advanced matching objectives to improve the training performance of distilled datasets, including matching training performance via meta learning [51, 7], matching feature regression performance [37, 38, 61, 32, 33], matching training gradients [60, 58, 20], matching training trajectories [2, 8, 4], and matching feature statistics [59, 50].

Although impressive results have been achieved and it has been demonstrated that neural networks trained by a synthetic dataset with even only 1 image per class can yield reasonable performance

---

[*]Corresponding Author.

37th Conference on Neural Information Processing Systems (NeurIPS 2023).

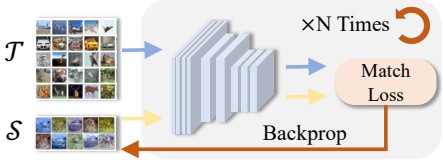 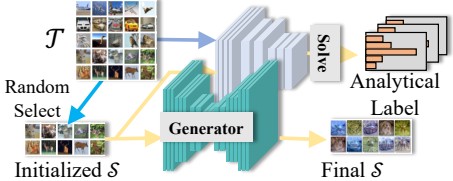

(a) Previous DD Paradigm        (b) MGDD Paradigm

Figure 1: Illustration of previous back-propagation-based and our generative fashions after adaptation for dataset distillation.

on real data, the iterative optimization process adopted in existing works results in significant computational overhead. As shown in Fig. 2, for FRePo [61], the method with the best performance requires above 3 hours to obtain a synthetic dataset with 1 image per class for CIFAR100 [23], let alone RTP [7], which is a back-propagation-through-time method and requires over 9 days for optimization. Such a dramatic latency makes existing approaches hard to be applied in scenarios requiring high efficiency, like handling data streams. Moreover, when the memory budget for a synthetic dataset changes, existing methods have to repeat the time-consuming optimization for the different sizes of $\mathcal{S}$, which lacks flexibility.

To alleviate the drawbacks of the conventional iterative forward-backward process, in this paper, we propose a generative dataset distillation approach, where the optimization loop is replaced by a single feed-forward propagation, as shown in Fig. 1(b). Specifically, given an initialization of synthetic samples in $\mathcal{S}$, we first obtain the corresponding synthetic labels by analytically solving a least-squares problem in a feature space. Then, a generator is adopted to transfer the initialized samples to the final ones. Our theoretical analysis indicates that $\mathcal{S}$ solved in an original space can be transferred to the final result by any conditional generator with an upper-bounded error, which validates the feasibility of this pipeline.

Then, the key problem of our framework lies in finding a suitable generator as quickly as possible for the feed-forward generation process of $\mathcal{S}$. To this end, we propose a method called *MGDD*, where a meta generator is learned with a meta learning algorithm on a large database like ImageNet [6]. Trained in a learning-to-learn fashion, the meta generator is optimized such that only a small number of adaptation steps are required for a target dataset unseen in the meta learning. Experiments demonstrate that our approach yields $22\times$ acceleration [2] and comparable performance with existing state-of-the-art methods, as shown in Fig. 2. Beyond that, the generator once adapted can also handle unseen sizes of $\mathcal{S}$ during adaptation, which improves the flexibility of cross-size generalization in existing DD methods significantly. We also validate that MGDD gets competitive performance on target datasets with large domain shifts from those seen in the meta learning.

Our contributions can be summarized from the following three aspects:

- We propose an innovative feed-forward generation fashion for DD without backward propagation after adaptation which significantly improves the efficiency of existing methods;

- We introduce MGDD which uses a meta-learning algorithm to learn a meta generator and helps the generator adapt to a target dataset rapidly;

- The proposed method achieves significant acceleration and improvement in the flexibility of cross-size generalization for existing DD approaches with comparable performance.

## 2 Related Works

Unlike conventional efficient learning schemes that mainly focus on lightening models [9, 34, 55, 54, 10, 18, 19, 17], dataset distillation (DD) looks into data compression: given a real large dataset $\mathcal{T}$, DD aims at a smaller synthetic dataset $\mathcal{S}$ which can match the training performance of $\mathcal{T}$. The seminal work by Wang *et al.* [51] proposes a meta learning approach to model this objective: in meta training, a network is trained with the current $\mathcal{S}$ for multiple times, while in meta test, the loss for the updated network is evaluated on $\mathcal{T}$, which is then back-propagated through the bi-level optimization

---

[2]The acceleration factor estimated here includes adaptation time. If only feed-forward time is considered, the acceleration can be $1650\times$.

to update $\mathcal{S}$. The following work by Deng *et al.* [7] improves performances by adopting momentum during meta training.

Considering the concerns on memory and time complexity of unrolling the computational graph in meta learning, a variety of works introduce various surrogate matching objectives to update $\mathcal{S}$. Zhao *et al.* [60] propose to match training gradients of $\mathcal{S}$ with those of $\mathcal{T}$, and following researches [20, 27, 16, 57, 45] focus on improving the classical gradient-matching objective. Beyond the single-step gradient, Cazenavette *et al.* [2] and subsequent works [4, 8] consider regulating multi-step training effects and propose matching training trajectories. Without the necessity of calculating higher order gradients, the distribution matching methods [59, 50] minimize the distance between feature statistics of $\mathcal{S}$ and $\mathcal{T}$, and result in satisfactory computational efficiency. Another branch of methods [37, 38, 61, 32, 33] turn to kernel ridge regression (KRR) model to improve the efficiency of the seminal meta learning based solution, since KRR enjoys the analytical form of solution, which gets rid of the meta-training process and yields best trade-off between performance and efficiency.

Different from works focusing on objectives of DD, some other researches explore methods of synthetic data parameterization, where synthetic samples can be stored in some other formats except the raw one to improve the data efficiency, and raw samples are recovered via some functions during down-stream training, *e.g.*, data augmentation [58], up-sampling [20], linear transformation [7], and neural networks [30, 26, 3, 49].

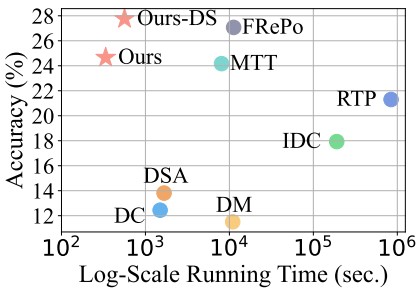

For a thorough introduction to dataset distillation, we refer readers to the recent surveys for this area [56, 13, 43, 28]. For existing methods, no matter what objectives and parameterizations are adopted, they all rely on an intensive loop of forward-backward propagation through a massive number of neural networks. Although a concurrent work [31] also adopts a similar pre-training and adaptation pipeline, it still relies on an iterative loop to solve an initialization of synthetic data.

Figure 2: Results of different DD methods on CIFAR100 with 1 image per class. Our MGDD achieves state-of-the-art efficiency and at least comparable performance. DS denotes down-sampling parameterization.

Different from prior works, we innovatively introduce a feed-forward fashion for dataset distillation in this paper. In fact, our method is orthogonal to different training objectives and data parameterizations, and can be built upon any combination of them. In this paper, we consider the KRR-based objective thanks to its favorable performance and computational efficiency and experiment with both raw and down-sampled parameterizations.

## 3   Methods

In this section, we elaborate on the technical methods of the proposed MGDD pipeline. We first introduce in Sec. 3.1 some preliminary information related to the matching objective. Then, for the main method, according to the overview of the whole pipeline in Fig. 1(b), the final synthetic labels and samples are derived by solving a least-squares problem and a conditional generator given initial synthetic data, which would be illustrated in detail in Sec. 3.2 and 3.3 respectively. Finally in Sec. 3.4, we analyze the feasibility of this two-step pipeline theoretically.

### 3.1   Preliminary

Let us denote datasets before and after distillation as $\mathcal{T} = (X_t, Y_t)$ and $\mathcal{S} = (X_s, Y_s)$ respectively, where $X_t \in \mathbb{R}^{n_t \times d}$, $Y_t \in \mathbb{R}^{n_t \times c}$, $X_s \in \mathbb{R}^{n_s \times d}$, $Y_s \in \mathbb{R}^{n_s \times c}$, $n_t$ and $n_s$ are number of real and synthetic data respectively, $d$ and $c$ are input and output dimensions respectively. Typically, for the RGB image classification task, $d$ is equal to $h \times w \times 3$, $c$ is the number of classes, and $Y_t$ is organized in the one-hot format. For the objective of DD, in this paper, we mainly consider the KRR-based methods in neural feature spaces [61, 32] due to its overall superior performance in terms of accuracy and efficiency. Specifically, assume there is an optimal neural network $f_{\theta^*}$ to projects $X_t$ and $X_s$ to a feature space with $p$ dimensions and $n_s \ll n_t < p$. The prediction error on $\mathcal{T}$ for the optimal KRR

parameter solved by $\mathcal{S}$, denoted as $W_s^{\theta^*}$ is adopted as the loss function:

$$\mathcal{L}(\mathcal{S};\theta^*) = \|f_{\theta^*}(X_t)W_s^{\theta^*} - Y_t\|_2^2 = \|f_{\theta^*}(X_t)f_{\theta^*}(X_s)^\top(f_{\theta^*}(X_s)f_{\theta^*}(X_s)^\top)^{-1}Y_s - Y_t\|_2^2. \quad (1)$$

In practice, since the optimal parameter $\theta^*$ is unknown, it is approximated by different random initializations [32] or alternately optimization with $\mathcal{S}$ [61].

## 3.2  Solving Synthetic Labels

Through Eq. 1, we can find that the loss in a neural space $\theta$ is upper-bounded by the distance between parameters solved by $\mathcal{S}$ and $\mathcal{T}$:

$$\begin{aligned}
\mathcal{L}(\mathcal{S};\theta) = \|f_\theta(X_t)W_s^{\theta^*} - Y_t\|_2^2 &= \|f_\theta(X_t)W_s^\theta - f_{\theta^*}(X_t)f_\theta(X_t)^\dagger Y_t\|_2^2 \\
&= \|f_\theta(X_t)W_s^\theta - f_\theta(X_t)W_t^\theta\|_2^2 \leq \|f_\theta(X_t)\|_2^2\|W_s^\theta - W_t^\theta\|_2^2 \quad (2) \\
&= \|f_\theta(X_t)\|_2^2\|f_\theta(X_s)^\dagger Y_s - W_t^\theta\|_2^2,
\end{aligned}$$

where $^\dagger$ denotes the pseudo-inverse of a matrix. In our MGDD framework, synthetic samples $X_s$ are initialized as some random real samples in $X_t$. Given a fixed $X_s$ and a random network $f_\theta$, the upper bound in Eq. 2 forms a least-squares problem with respect to synthetic labels $Y_s$, which can be minimized by an analytically optimal solution:

$$Y_s^* = f_\theta(X_s)W_t^\theta = f_\theta(X_s)f_\theta(X_t)^\top(f_\theta(X_t)f_\theta(X_t)^\top)^{-1}Y_t. \quad (3)$$

$Y_s^*$ obtained with Eq. 3 serves as final synthetic labels. In the next subsection, we will introduce the generation of synthetic samples conditioned on their initialization.

## 3.3  Learning a Synthetic Sample Generator

Conditioned on initialized synthetic samples $X_s$, a generator $g_\omega$ is adopted to predict the final synthetic data $X_s^*$, where the parameter $\omega$ can encode useful information of the target dataset $\mathcal{T}$ and the optimal neural space parameterized by $\theta^*$. We expect that the generator can acquire such knowledge through a fast learning process within a limited number of training steps. To this end, we propose a learning-to-learn algorithm based on MAML [12], where a meta generation network is learned to optimize the performance of the network adapted for a few steps from the meta one.

Specifically, to ensure the generality of the meta generator for different target datasets, we perform the training algorithm on ImageNet1k [6], a large-scale dataset for image classification. In each training iteration, a subset of all classes is randomly sampled from it to mimic different target datasets that the generator may encounter in practice. And the meta generator is learned in a bi-level learning framework including a meta-training loop and a meta-testing step, and the meta-testing loss is back-propagated through the computational graph of meta-training steps to update the parameter of the meta generator.

In each meta-training and meta-testing step, from the selected classes, we randomly sample a batch of real images as $\mathcal{T}$ and initialize synthetic data $X_s$ with part of them. With a random and fixed neural network as $f_\theta$, the synthetic labels $Y_s^*$ are solved via Eq. 3. Then, the final synthetic samples $X_s^*$ are predicted by the current generator in a forward pass, and $\mathcal{S} = (X_s^*, Y_s^*)$ is evaluated by the loss $\mathcal{L}(\mathcal{S};\theta^*)$ in Eq. 1. In this paper, following Loo *et al.* [32], we approximate the optimal neural parameter $\theta^*$ via random sampling in different steps from the distribution for initialization. The loss signal is back-propagated to the current generator and the meta generator to update parameters in meta-training and meta-testing respectively. It is worth noting that in different meta-training and meta-testing times, we use different sizes of synthetic data, which enhances the cross-size generalization ability on target datasets since the meta-testing losses on sizes probably unseen during meta-training are optimized. The main algorithm is summarized in Alg. 1. Given a trained meta generator, a limited number of adaptation steps are performed for a target dataset. The procedure of adaptation is similar to the meta-training loop in Alg. 1.

As for the architecture of the generator, in this paper, we adopt a simple encoder-decoder model, where the encoder consists of three convolutional blocks with two average pooling layers while the decoder has a symmetric structure. Notably, we observe in practice that it is beneficial for different sizes of synthetic datasets to adopt different transfer functions. Taking various sizes into consideration, we concatenate additional size-embedding channels to the bottle-necked layer of the

**Algorithm 1** MGDD: Meta-Learning Framework for Training a Synthetic Sample Generator

---

**Input:** $\mathcal{Z}$: A Large Dataset; $\theta$: Parameter of a Random Neural Network; $T$: Number of Meta-Training Steps; $\alpha$: Learning Rate in Meta-Training; $\beta$: Learning Rate in Meta-Testing.
**Output:** $\omega$: Parameter of a Meta Generator.

 1: **procedure** GETTRAININGLOSS($\mathcal{T}$)
 2:     Initialize $X_s$ with some random real images from $\mathcal{T}$;
 3:     Obtain $Y_s^*$ with the analytical solution in Eq. 3;
 4:     Forward propagation with $X_s^* \leftarrow g_{\omega'}(X_s)$;
 5:     Randomly sample $\theta^*$ and compute the loss $\mathcal{L}((X_s^*, Y_s^*); \theta^*)$ in Eq. 1;
 6:     **return** $\mathcal{L}((X_s^*, Y_s^*); \theta^*)$
 7: **end procedure**
 8: Initialize $\omega$ randomly;
 9: **repeat**
10:     $\omega' \leftarrow \text{copy}(\omega)$;
11:     Randomly choose a subset of classes $\mathcal{C}$ from $\mathcal{Z}$;
12:     Sample a batch of images of $\mathcal{C}$ as $\mathcal{T}$;
13:     **for** $1 \leq i \leq T$ **do**
14:         $\mathcal{L} = \text{GetTrainingLoss}(\mathcal{T})$;                          ▷ Meta-Training
15:         Back propagation and update $\omega'$: $\omega' \leftarrow \omega' - \alpha \nabla_{\omega'} \mathcal{L}$;
16:     **end for**
17:     $\mathcal{L} = \text{GetTrainingLoss}(\mathcal{T})$;                              ▷ Meta-Testing
18:     Back propagation and update $\omega$: $\omega \leftarrow \omega - \beta \nabla_{\omega} \mathcal{L}$;
19: **until** convergence

---

generator, inspired by the design of the position embedding in Transformer models [47] and the time-step embedding in diffusion models [15, 39, 41]. Please refer to the appendix for detailed architecture configurations.

### 3.4 Theoretical Analysis

There are two key steps in the proposed MGDD framework: solving the optimal synthetic labels $Y_s^*$ as introduced in Sec. 3.2 and generating the corresponding synthetic samples $X_s^*$ as introduced in Sec. 3.3. Define the error $\mathcal{L}(\mathcal{S}; \theta)$ in Eq. 1 using the fixed $X_s$ and the optimal $Y_s$ in Eq. 3 with the projection function of $f_\theta$ as $\epsilon$. The reason we pursue the optimal $Y_s$ is that the final $X_s^*$ is transferred from the initial $X_s$, whose error in the optimal neural space parameterized by $\theta^*$ is upper-bounded by $\epsilon$, as derived in the following theorem.

**Theorem 1.** *Given $f_\theta : \mathbb{R}^d \to \mathbb{R}^p$, $f_{\theta^*} : \mathbb{R}^d \to \mathbb{R}^p$, $X_t \in \mathbb{R}^{n_t \times d}$, $Y_t \in \mathbb{R}^{n_t \times c}$, $X_s \in \mathbb{R}^{n_s \times c}$, $d < p$, $Y_s^*$ obtained by Eq. 3 with the optimal $\mathcal{L}((X_s, Y_s^*); \theta)$ denoted as $\epsilon$, and an arbitrary transfer function $g_\omega$ parameterized by $\omega$ taking $X_s$ as input, the transferred $g_\omega(X_s)$ yields an upper-bounded loss $\mathcal{L}((g_\omega(X_s), Y_s^*); \theta^*)$.*

*Proof.* We first rewrite the given condition:

$$\mathcal{L}((X_s, Y_s^*); \theta) = \|f_\theta(X_t)W_s^\theta - Y_t\|_2^2 = \|f_\theta(X_t)f_\theta(X_s)^\top(f_\theta(X_s)f_\theta(X_s)^\top)^{-1}Y_s^* - Y_t\|_2^2 = \epsilon. \tag{4}$$

Then, we have:

$$\begin{aligned}
\mathcal{L}((g_\omega(X_s), Y_s^*); \theta^*) &= \|f_{\theta^*}(X_t)W_s^{\theta^*} - Y_t\|_2^2 \\
&= \|f_{\theta^*}(X_t)f_{\theta^*}(g_\omega(X_s))^\top(f_{\theta^*}(g_\omega(X_s))f_{\theta^*}(g_\omega(X_s))^\top)^{-1}Y_s^* - Y_t\|_2^2 \\
&\leq \|f_{\theta^*}(X_t)W_s^{\theta^*} - f_\theta(X_t)W_s^\theta\|_2^2 + \|f_\theta(X_t)W_s^\theta - Y_t\|_2^2 \\
&= \|f_{\theta^*}(X_t)W_s^{\theta^*} - f_\theta(X_t)W_s^\theta\|_2^2 + \epsilon,
\end{aligned} \tag{5}$$

where the inequality is based on the triangle inequality and the last equation is due to Eq. 4. □

Theorem 1 indicates that to achieve feed-forward dataset distillation, we do not need to design a neural network taking the whole original dataset $\mathcal{T}$ as input. We can in fact adopt a conditional

| Dataset | CIFAR10 | | | CIFAR100 | |
|---|---|---|---|---|---|
| IPC | 1 | 10 | 50 | 1 | 10 |
| Ratio (%) | 0.02 | 0.2 | 1 | 0.2 | 1 |
| Random     Acc. (%) | 21.87±0.30 | 38.86±0.31 | 57.07±0.54 | 7.06±0.16 | 24.50±0.08 |
| Full Dataset   Acc. (%) | | 84.8±0.1 | | | 56.2±0.3 | |
| DC [60]    Acc. (%) | 28.20±0.71 | 43.74±0.41 | 53.43±0.28 | 12.44±0.18 | 25.08±0.17 |
| Time (sec.) | 153 | 3605 | 20090 | 1496 | 33749 |
| DSA [58]    Acc. (%) | 28.10±0.72 | 52.15±0.48 | 60.58±0.29 | 13.81±0.21 | 32.49±0.30 |
| Time (sec.) | 172 | 3871 | 21217 | 1667 | 35905 |
| IDC [20]    Acc. (%) | 35.34±0.87 | 58.50±0.39 | 69.32±0.30 | 17.93±0.15 | 36.08±0.38 |
| Time (sec.) | 39062 | 40244 | 45888 | 189811 | 198101 |
| MTT [2]    Acc. (%) | 45.29±0.86 | 62.77±0.56 | 71.09±0.34 | 24.17±0.57 | 39.43±0.26 |
| Time (sec.) | 2972 | 8588 | 8601 | 8063 | 10295 |
| DM [59]    Acc. (%) | 27.08±0.36 | 48.80±0.31 | 62.94±0.28 | 11.51±0.25 | 29.33±0.23 |
| Time (sec.) | 1123 | 1182 | 1423 | 10942 | 11355 |
| RTP [7]    Acc. (%) | 49.1±0.6* | 62.4±0.4* | 70.5±0.4* | 21.3±0.6* | 34.7±0.5* |
| Time (sec.) | 816212 | 1647559 | 1660726 | 842801 | 1672205 |
| FRePo [61]    Acc. (%) | 43.24±0.32 | 65.76±0.72 | 71.03±0.34 | 27.07±0.26 | 39.97±0.32 |
| Time (sec.) | 8507 | 11133 | 29185 | 11231 | 37607 |
| Ours    Acc. (%) | 46.26±0.27 | 60.76±0.38 | 69.50±0.17 | 24.66±0.15 | 36.47±0.27 |
| Time (sec.) | **120**×6802 | **505**×22 | **1395**×6 | **505**×22 | **3004**×13 |
| Label (sec.) | 7 | 7 | 7 | 7 | 7 |
| Generator (sec.) | 113 | 498 | 1388 | 498 | 2997 |
| Forward (ms) | 2 | 3 | 11 | 3 | 20 |

Table 1: Comparisons on test accuracy and running time with state of the arts. The acceleration marked by the red subscript is computed against the method with the best accuracy. We also provide detailed analysis for our method on the time cost for each component in adaptation, including solving synthetic labels, updating the generator from the meta generator, and feed-forward generation. IPC: Number of Images Per Class; Ratio: ratio of distilled images to the whole training set. * denotes results from the original paper.

generation function that transfers the initial synthetic samples to the desired ones, which has an error upper bound as shown in Theorem 1. Since the upper bound is related to the error in the original space parameterized by $\theta$, it is crucial to solve the optimal synthetic labels with respect to $\theta$ in the first step. Also, we notice from Eq. 5 that the optimal generator is dependent on $\mathcal{T}$ and $\theta^*$. Given that $\theta^*$ is intractable and can only be approximated by iterative sampling, we build a meta learning algorithm in Alg. 1 for the MGDD framework to enforce an efficient process to model this dependency in only a few steps.

## 4  Experiments

### 4.1  Implementing Details

As discussed in the previous section, there are 3 stages in the proposed MGDD, including training, adaptation, and inference stages. In the training stage, we aim at a meta generator $g_\omega$ and adopt Alg. 1 to train $g_\omega$ on a large dataset. In this paper, to ensure that the meta generator can acquire general knowledge for fast adaptation on a target dataset, we use ImageNet1k [6], a large-scale image recognition dataset popular in the computer vision and machine learning communities, as the dataset for meta learning. There are roughly 1,280,000 images in a total of 1,000 classes. We resize all images to the $32 \times 32$ resolution. In each outer loop of Alg. 1, we randomly select 100 classes at most and a batch of data with 2,000 images in maximal from the selected class as a current target dataset $\mathcal{T}$, with which we initialize a synthetic dataset $\mathcal{S}$ with 1,000 samples at most in each inner step. The training objective is based on Eq. 1 and the implementation follows the open-source code of FRePo [61]. For computational efficiency, the generator processes each sample independently and the configuration of the architecture can be found in the appendix. The meta generator is trained by the Adam optimizer [21] and the learning rate $\beta$ is set as $10^{-5}$. The learning rate in meta-training is set as $10^{-4}$ and the number of meta-training steps $T$ is 5. The meta generator is trained with 200,000 meta-testing iterations. We use a cloud server with a single A40 GPU for meta learning and a workstation with a single 3090 GPU for the subsequent adaptation. The meta learning takes roughly 2 days while the time cost of adaptation is analyzed in Tab. 1.

| IPC | Steps | Methods | P | A | C | S | PACS | Path | Blood | PathBlood | CUB200 |
|---|---|---|---|---|---|---|---|---|---|---|---|
| | 1,000 | Baseline | 53.54±0.99 | 33.53±3.19 | 51.69±1.25 | 39.76±1.74 | 38.03±0.60 | 54.23±3.12 | 68.38±0.55 | 55.11±1.26 | 4.84±0.19 |
| | | w/o Meta | 51.74±1.71 | 40.29±0.68 | 60.45±0.32 | 35.87±0.07 | 42.73±0.25 | 57.06±1.36 | 69.14±1.86 | 58.03±0.51 | 6.73±0.13 |
| | | Ours | **57.31±1.41** | **41.18±0.48** | **60.71±1.02** | **43.98±0.55** | **43.46±0.89** | **57.60±1.19** | **70.83±0.81** | **59.86±0.66** | **7.40±0.20** |
| 1 | 2,000 | Baseline | 55.80±1.16 | 40.87±1.47 | 56.40±2.57 | 44.26±0.52 | 38.53±0.80 | 54.70±1.07 | 68.55±1.39 | 60.62±1.14 | 5.65±0.26 |
| | | w/o Meta | 57.37±1.38 | 42.15±0.89 | 62.84±1.19 | 41.47±1.05 | 43.89±0.62 | 58.33±1.76 | 71.28±1.61 | 60.72±0.17 | 7.25±0.09 |
| | | Ours | **60.18±1.99** | **42.50±0.09** | **63.96±0.80** | **46.47±0.96** | **44.92±0.48** | **59.06±1.02** | **72.72±0.90** | **62.22±0.27** | **7.68±0.18** |
| | Full | Baseline | **68.64±0.52** | 41.59±1.53 | 59.38±0.93 | 52.85±0.51 | 44.87±0.28 | 64.74±1.10 | 71.17±0.72 | 64.57±1.63 | 12.41±0.20 |
| | | Ours | 66.77±1.81 | **44.24±0.36** | **64.48±0.68** | **53.40±1.58** | **48.39±0.49** | **70.25±0.11** | **72.93±1.70** | **65.30±0.44** | **12.51±0.27** |
| | 1,000 | Baseline | 76.33±0.31 | 47.20±1.61 | 70.92±1.02 | 63.26±1.26 | 59.54±0.34 | 76.18±0.87 | 80.09±0.82 | 74.46±0.91 | 10.33±0.64 |
| | | w/o Meta | 77.28±0.44 | 54.57±0.75 | 72.81±0.10 | 69.11±0.17 | 58.33±0.37 | 75.00±0.74 | 79.26±0.17 | 74.45±0.39 | 13.69±0.09 |
| | | Ours | **79.40±0.29** | **56.18±0.34** | **74.49±0.35** | **70.63±0.33** | **61.41±0.13** | **76.21±0.10** | **80.20±0.41** | **75.78±0.69** | **13.98±0.09** |
| 10 | 2,000 | Baseline | 77.59±0.41 | 51.18±2.31 | 72.36±0.44 | 67.68±0.86 | 59.86±0.21 | 76.96±0.70 | 81.73±0.37 | 75.01±0.51 | 11.25±0.51 |
| | | w/o Meta | 78.24±0.42 | 55.52±0.63 | 74.89±0.23 | 71.09±0.63 | 60.69±0.33 | 77.06±0.18 | 81.71±0.37 | 75.26±0.07 | 13.83±0.24 |
| | | Ours | **80.78±1.05** | **56.82±0.51** | **76.12±0.63** | **72.11±0.48** | **62.33±0.15** | **77.44±0.11** | **82.31±0.45** | **76.32±0.14** | **14.69±0.37** |
| | Full | Baseline | **85.40±0.30** | **60.64±1.64** | 78.10±0.54 | 76.87±1.09 | 65.38±0.48 | 77.44±1.53 | **85.81±0.56** | 76.58±0.25 | **16.84±0.12** |
| | | Ours | 82.27±0.89 | 59.47±1.20 | **78.88±0.15** | **76.91±0.25** | **65.76±0.26** | **78.21±1.42** | 84.47±0.23 | **77.07±0.98** | 15.05±0.37 |

Table 2: Evaluations on PACS, PathMNIST, BloodMNIST, and CUB200 datasets. The baseline is FRePo [61]. *w/o Meta* denotes adapting from scratch instead of a meta generator.

In the adaptation stage, we load the meta generator obtained by the training stage and try to adapt it for a target dataset. The optimization is similar to that in the meta-training step of Alg. 1 and the difference is that we use the Adam optimizer to update the parameters instead of pure gradient decent. Notably, there are two ways to initialize $X_s$ in each adaptation step, namely single-initialization and multi-initialization modes. In the single-initialization mode, $X_s$ is set as the same group of random real samples in the real dataset for all adaptation steps, while in the multi-initialization mode, it can be set differently, with various random seed or/and various sizes of synthetic datasets. We experiment with both ways in this section.

## 4.2 Comparisons with State of the Arts

In this part, we compare the proposed MGDD with existing state-of-the-art DD methods on standard benchmarks, including CIFAR10 and CIFAR100 [23] datasets. There are 50,000 training images and 10,000 testing images in the $32 \times 32$ resolution for both datasets, and the numbers of classes are 10 and 100 respectively. For CIFAR10, we compare our method with state of the arts on settings of 1, 10, and 50 images per class (IPC) in synthetic datasets and for CIFAR100, IPCs are 1 and 10. The candidates are three gradient-matching methods DC [60], DSA [58], and IDC [20], one trajectory-matching method MTT [2], one distribution-matching method DM [59], one back-propagation-through-time (BPTT) method RTP [7], and one kernel-ridge-regression-based method FRePo [61]. We obtain synthetic datasets using each algorithm on a 3-layer `ConvNet` whose architecture is the same as that in [61] and evaluate the accuracy on testing datasets for networks with the same architecture trained on synthetic datasets. For our method, we adapt the generator from the meta model on each comparison setting for 10,000 steps, and for each setting, we report mean and standard deviation over the accuracy of 3 networks initialized from scratch. Moreover, we report the cost of time to derive synthetic datasets.

As shown in Tab. 1, our method achieves comparable performance with DD methods based on heavy iterative loops and yields significant acceleration. For gradient-matching-based based solutions, they typically rely on computing second-order derivatives over a large amount of neural networks to back-propagate the matching loss to synthetic data, which results in high computation overhead. Even worse, for trajectory-matching and BPTT methods, higher-order derivatives are necessary. As a result, their time cost for optimizing synthetic datasets is formidable, and our method achieves up to $6800\times$ acceleration compared with BPTT. For DM and FRePo, although they do not rely on high-order gradients, they typically require a massive number of forward-backward iterations to update synthetic datasets, *e.g.*, the FRePo baseline requires 500,000 iterations for optimization, and the efficiency is still unsatisfactory. On the contrary, the results of our method are produced by a generator in a one-stop fashion, which is trained by a meta-learning algorithm and learns to adapt to a target dataset rapidly. It thus enjoys the superior running efficiency of DD. More analysis of the accuracy performance given the same running time can be found in the appendix.

Notably, although the generator is only trained under $32 \times 32$ RGB images in meta-learning, given that it has a fully-convolutional structure and maintains the same resolution for input and output, it

| Dataset | CIFAR10 | | | CIFAR100 | |
|---|---|---|---|---|---|
| IPC | 1 | 10 | 50 | 1 | 10 |
| FRePo | 26.8±0.7 | 49.6±0.1 | 62.0±0.4 | 10.1±0.3 | 29.6±0.2 |
| Ours w FRePo | 42.6±0.3 | 58.9±0.4 | 66.8±0.2 | 20.8±0.2 | 32.2±0.3 |
| DC | 24.7±0.4 | 43.1±0.3 | 56.0±0.3 | 6.6±0.2 | 19.8±0.3 |
| Ours w DC | 34.3±0.4 | 46.2±0.6 | 60.0±0.4 | 14.1±0.1 | 22.3±0.4 |
| DM | 25.7±0.5 | 45.5±0.4 | 57.4±0.5 | 9.6±0.2 | 21.9±0.3 |
| Ours w DM | 31.4±0.2 | 48.9±0.2 | 60.8±0.4 | 16.6±0.2 | 22.9±0.3 |

Table 3: Performance of using different DD training objectives during adaptation.

| IPC | Method | Train Arch. ConvNet | Unseen Arch. AlexNet | VGG | ResNet |
|---|---|---|---|---|---|
| 1 | Baseline | 49.6±0.1 | 44.5±0.7 | 33.0±0.1 | 31.8±1.6 |
| | Ours | 58.9±0.4 | 55.1±0.4 | 35.9±0.6 | 32.7±0.8 |
| 10 | Baseline | 26.8±0.7 | 23.4±0.3 | 16.9±0.1 | 15.1±0.8 |
| | Ours | 42.6±0.3 | 39.6±0.8 | 22.9±0.6 | 19.1±1.3 |
| 50 | Baseline | 62.0±0.4 | 59.2±0.3 | 48.7±1.1 | 48.2±0.4 |
| | Ours | 66.8±0.2 | 62.8±0.2 | 50.9±0.7 | 52.4±1.2 |

Table 4: Performance of different network architectures on CIFAR10.

is applicable to datasets with larger resolutions. It can also be adapted for datasets with different numbers of input channels with minor modifications. In the appendix, we provide more experimental results of such cross-resolution and cross-channel-number generalization cases.

### 4.3 Empirical Studies

In this part, we focus on some interesting properties of the proposed MGDD method, including cross-dataset, cross-objective, cross-architecture, cross-synthetic-initialization, cross-parameterization, and cross-IPC settings. More studies including cross-resolution, cross-channel-number, and cross-class-number that cannot fit into the main content are put in the appendix.

**Cross-Dataset Generalization:** MGDD proposed in this paper is expected to be generalized to any downstream target datasets, including those unseen and with large domain shifts from datasets used in the meta learning. To evaluate the cross-dataset generalization performance of MGDD, we conduct experiments on more datasets including one domain generalization dataset PACS [29], two medical classification datasets PathMNIST and BloodMNIST [53], and one fine-grain image classification dataset CUB200 [48]. PACS contains 9,991 images from 7 classes and 4 domains: Photo (P), Art Painting (A), Cartoon (C), and Sketch (S). The style variations across the 4 domains are drastic. We perform dataset distillation on each domain both independently and jointly, which formulates a 28-class dataset. PathMNIST and BloodMNIST contain 107,180 images of 9 classes and 17,092 images from 8 classes respectively. We also combine them together to form a 17-class dataset denoted as PathBloodMNIST. CUB200 contains 6,000 images of 200 bird species. We process all the images to the $32 \times 32$ resolution in the RGB format and compare the performance with the FRePo baseline [61] and a generator from scratch instead of the meta generator, under 1,000 and 2,000 steps as well as full convergence. The quantitative results shown in Tab. 2 validate the robustness of MGDD under various datasets and domain shifts.

**Cross-Objective Generalization:** By default, both the meta-learning and adaptation objectives used in this paper for MGDD are the KRR objective in Eq. 1 following FRePo [61]. Empirically, we find that it is also feasible to adopt different objectives for adaptation. Here, we switch the adaptation objective to DC [60] and DM [59] respectively. The optimization steps for both baselines and our method are set as 2,000. As shown in Tab. 3, our methods yield consistent improvement over different baselines in a limited number of optimization steps.

**Cross-Architecture Generalization:** Adapted on an original architecture, a satisfactory generator is expected to produce results that are also valid to train networks with different structures, namely cross-architecture generalization. Tab. 4 shows the performance on CIFAR10, where ConvNet is used in adaptation while AlexNet [24], VGG11 [46], and ResNet18 [14] are used as unseen architectures for evaluation. The results indicate that the accuracy improvement on the original model still holds for unseen structures.

**Cross-Initialization Generalization:** We find that the single-initialization scheme used in Tab. 1 may lead to over-fitting of the generator to the adopted single initialization. As shown in Tab. 5, if we change the initialization of synthetic data, the performance would drop dramatically. Fortunately, multi-initialization is an alternative to account for this drawback, wherein each adaptation step, a new initialization of the synthetic dataset is sampled from the original dataset. Tab. 5 indicates that multi-initialization typically requires more adaptation steps for convergence and can perform on par with single-initialization. It is useful when samples in the synthetic dataset are concerned with user privacy, given that visualization results produced by the KRR objective are somehow realistic, as illustrated in Fig. 6 and [61]. In such cases, replacing the current dataset with the results of another

| Setting | Default | | Multi-Init |
|---|---|---|---|
| | Original | Different Init | Different Init |
| Acc. (%) | 60.76±0.38 | 43.27±0.46 | 60.24±1.28 |
| Time (sec.) | 505 | | 1108 |

| Dataset | CIFAR10 | | CIFAR100 |
|---|---|---|---|
| IPC | 1 | 10 | 1 |
| Ours | 46.26±0.27 | 60.76±0.38 | 24.66±0.15 |
| Ours-DS | 48.17±0.77 | 63.97±0.30 | 27.76±0.44 |

Table 5: Using multiple initialized samples during adaptation can enhance the cross-initialization performance. Performance on CIFAR10 with 10 IPC is shown here.

Table 6: Our method is orthogonal to synthetic dataset parameterization methods and provides improvement gain on small synthetic datasets. DS denotes down-sampling parameterization.

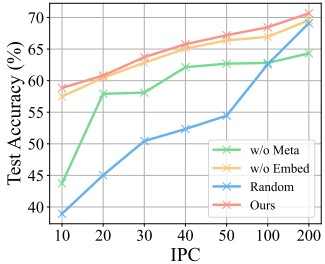

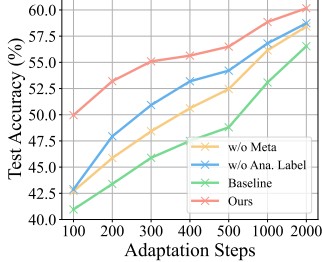

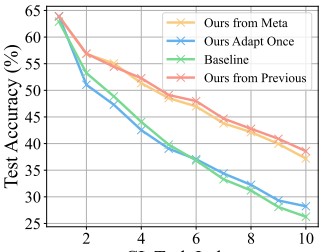

Figure 3: Performance of various IPCs on CIFAR10. Only 10 and 50 are seen in adaptation.

Figure 4: Performance of generators with various adaptation steps on CIFAR10 with 10 IPC.

Figure 5: Performance of continual learning on CIFAR100.

initialization would help solve the problem efficiently, without the necessity to re-run the whole optimization loop of existing methods.

**Cross-Parameterization Generalization:** Beyond different training objectives, the proposed MGDD is also orthogonal with different synthetic data parameterization tricks. In Tab. 6, we consider storing $2\times$ down-sampled synthetic images instead of the raw ones. Thus, $4\times$ synthetic samples can be stored given the same storage budget. We find that the simple strategy can lead to additional performance gain for relatively small budgets. The observation is consistent with previous works [20, 30, 7].

**Cross-IPC Generalization:** One crucial benefit of the MGDD is the cross-IPC generalization. Once the generator is adapted for a target dataset, when the storage budget changes, we do not need to perform the optimization again, unlike previous solutions relying on iteratively updating synthetic datasets. To demonstrate the cross-IPC generalization performance, we conduct experiments on CIFAR10 and adapt the meta generator using multi-initialization, with 10 and 50 IPCs for 2,000 steps. The adapted generator is evaluated on unseen IPCs 20, 30, and 40. The results are shown in the red curve of Fig. 3, where the generator produces satisfactory synthetic datasets with unseen IPCs.

To make the generator aware of the sizes of synthetic datasets, we concatenate size embedding channels to features at the middle layer of the generator. To understand how the embedding works, we remove these channels and conduct the same evaluation. As shown in the yellow curve of Fig. 3, the performance degrades without size embedding. In Fig. 6, we visualize some samples before and after the adapted generator on CIFAR10 with 1 and 50 IPCs. We can observe that the results are quite different: for small sizes, the generated results are vague, while for large sizes the results are more realistic, and their styles are also different. Thus, it is reasonable to take different sizes of synthetic datasets into consideration in the inference stage.

We also try training the generator from scratch instead of the meta model on the target dataset. As shown in Fig. 3, the worse performance in the green curve indicates that meta-learning is crucial for finding a satisfactory initial checkpoint for downstream adaptation.

Moreover, we also evaluate the performance on higher IPCs like 100 and 200, and the results are still encouraging compared with random real samples, which indicates that our method can serve as an alternative when the computational resource cannot support optimization for larger IPCs directly.

**Various Adaptation Steps:** In Fig. 4, we visualize the accuracy under different adaptation steps on CIFAR10 with 10 IPC as the red curve. Compared with training from scratch and the baseline FRePo [61], as shown in the yellow and green curves respectively, our method results in significantly faster adaptation convergence, which would be attributed to the good initial generator found by the meta-learning algorithm. Thus, our method is more applicable in scenarios requiring high efficiency, like processing streaming data.

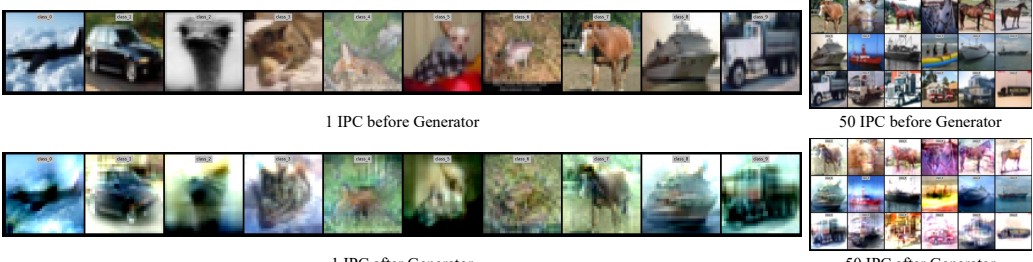

Figure 6: Visualizations of samples before and after generator on CIFAR10 with 1 and 50 IPC.

Furthermore, to demonstrate the effectiveness of analytical labels, we replace them with vanilla one-hot labels in synthetic datasets and the performance is shown in the blue curve. The considerable performance drop indicates the importance of minimizing the error in an original space via analytical labels, which is consistent with the theoretical analysis in Theorem 1.

### 4.4 Application: Continual Learning

Continual learning (CL) aims to learn from a stream of data, where historical and future data are unavailable when learning the current data batch. One important issue is catastrophic forgetting [22]: a model tends to forget knowledge acquired in previous data when learning on newly-coming data. Focusing on this drawback, many works introduce a buffer with limited memory to store core data and knowledge of past experience for future use [40, 1]. Dataset distillation benefits this field by generating informative samples [36, 5, 42, 35, 44] to prevent forgetting as much as possible.

In this paper, we evaluate the CL performance of the proposed MGDD on CIFAR100, following the same protocol of [61, 59], where all 100 classes are divided into 10 tasks randomly with 10 classes for each task. For each task, a buffer with 20 images for each class is allowed for synthetic data. We first try adapting the generator on each new task from the meta model for 2,000 steps and the performance is shown in the yellow curve in Fig. 5. Alternatively, we can choose to adapt the generator from the checkpoint of the previous task, which has already learned some global knowledge of full data and yields better performance, as shown in the red curve. Notably, it is also feasible to only adapt the generator on the first task and the remaining tasks directly adopt this generator to output synthetic data. As shown in the blue curve, with the most significant flexibility, the performance is still comparable with the FRePo baseline [61] shown in the green curve, which suggests great practical value for processing data streams.

## 5 Conclusion

In this paper, we propose MGDD, a novel feed-forward paradigm for dataset distillation (DD). Specifically, in our pipeline, synthetic labels are obtained by solving a least-squares problem equipped with an analytical solution, and synthetic samples are transferred from their initial results by a conditional generator instead of taking the whole original training dataset as input. Theoretical derivation indicates an error upper bound of the proposed framework. On the one hand, unlike existing DD approaches requiring time-consuming forward-backward iterations through a massive number of networks, MGDD generates distilled results with a generator adapted rapidly from a meta generator, which improves the efficiency of DD significantly. On the other hand, existing techniques have to repeat the whole iterative algorithms for different sizes of synthetic datasets, while MGDD can perform inference flexibly on various sizes once adapted to the target dataset. Focusing on the efficiency of adaptation on target datasets, we propose a meta-learning algorithm to train a meta generator, such that it can acquire knowledge of target datasets sufficiently in only a few steps. Experiments demonstrate that the proposed MGDD performs on par with existing state-of-the-art DD baselines under $22\times$ acceleration. It also exerts strong cross-size generalization ability even on sizes of synthetic datasets unseen during adaptation. Future works may explore advanced feed-forward fashions of DD, focusing on generation pipelines, training algorithms, and network architectures, making improvements on the cross-dataset, cross-size, and cross-architecture generalization.

## Acknowledgment

This work is supported by the Advanced Research and Technology Innovation Centre (ARTIC), the National University of Singapore under Grant (project number: A0005947-21-00, project reference: ECT-RP2), and the Singapore Ministry of Education Academic Research Fund Tier 1 (WBS: A0009440-01-00).

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

In this part, we include more details about the technical methods, more experimental results of our MGDD method, and more discussion on limitations and future works, which cannot be accommodated in the main paper due to the page limit. Our method first trains a meta generator to generate synthetic samples and then an adaptation stage is executed for a target dataset. We provide algorithmic details of the adaptation stage, a summary of hyper-parameters, and configurations of our generator architecture. Then, we conduct more evaluations on the cross-number-of-channel, cross-resolution, cross-ipc, and cross-number-of-classes performance of our method. More discussions of the adaptation performance and more qualitative examples are also included. Finally, we discuss limitations of the proposed method and potential future works.

## A   More Details

**Adaptation Algorithm:** Alg. 1 demonstrates the procedure of meta learning to obtain a meta synthetic sample generator. On a downstream target dataset, the meta network is adapted to a specific network with a limited number of steps. The adaptation algorithm is similar to the meta-training step of the meta learning algorithm. Here, we present the full details in Alg. 2. We can prepare multiple initialization of synthetic samples through randomly sampling from the target dataset. Recall that the main pipeline of our algorithm is to first obtain analytical synthetic labels in a random neural space $\theta$: $Y_s^* = f_\theta(X_s)W_t^\theta$. Here, the optimal kernel-ridge-regression parameters of the target dataset $W_t^\theta$ can be computed by $W_t^\theta = f_\theta(X_t)^\top (f_\theta(X_t)f_\theta(X_t)^\top)^{-1}Y_t$, if the number of real samples $n_t$ is smaller than the feature dimension $p$. Otherwise, $W_t^\theta = (f_\theta(X_t)^\top f_\theta(X_t))^{-1}f_\theta(X_t)^\top Y_t$.

---

**Algorithm 2** Adaptation Algorithm of Synthetic Sample Generator for a Target Dataset

---

**Input:** $(X_t, Y_t)$: A Target Dataset; $T$: Number of Adaptation Steps; $\alpha$: Learning Rate of Generator; $\theta$: Parameter of a Random Neural Network; $\omega$: Parameter of a Meta Generator; $\mathcal{I}$: A Set of Randomly Initialized Synthetic Samples.
**Output:** $\omega'$: Parameter of a Target-Specific Generator.
1: $W_t^\theta = f_\theta(X_t)^\top (f_\theta(X_t)f_\theta(X_t)^\top)^{-1}Y_t$;
2: **for** Each $X_s$ in $\mathcal{I}$ **do**
3:     $Y_s^* = f_\theta(X_s)W_t^\theta$;                                              ▷ Eq. 3
4: **end for**
5: Initialize generator parameters $\omega'$ with $\omega$;
6: **for** $1 \le i \le T$ **do**
7:     Sample a batch of real data $(X_t^i, Y_t^i)$ of from $(X_t, Y_t)$;
8:     Sample a initialized synthetic data $(X_s, Y_s^*)$ from $\mathcal{I}$;
9:     $X_s^* = g_{\omega'}(X_s)$;                                     ▷ Forward propagation
10:    Sample neural parameters $\theta^*$ from a random distribution;
11:    $\mathcal{L} = \|f_{\theta^*}(X_t)f_{\theta^*}(X_s^*)^\top (f_{\theta^*}(X_s^*)f_{\theta^*}(X_s^*)^\top)^{-1}Y_s^* - Y_t\|_2^2$;          ▷ Eq. 1
12:    Update $\omega'$ via $\omega' \leftarrow \omega' - \alpha\nabla_{\omega'}\mathcal{L}$;              ▷ Back propagation
13: **end for**

---

After the calculation of analytical labels, we fix them and train the synthetic sample generator initialized by parameters of the meta generator for some steps. The optimization objective is similar to those in Zhou *et al.* [61] and Loo *et al.* [32]. The difference is that the optimization target is parameters of the generator instead of synthetic samples.

**Summary of Hyper-Parameters:** For a clear view, we summarize the hyper-parameters and their values in both meta learning and adaptation stages as shown in Tab. 7. All experiments follow these default settings of hyper-parameters if not specified. Other configurations unmentioned follow the settings of the baseline FRePo [61].

**Generator Architecture:** We illustrate the detailed configurations of our generator architecture in Fig. 7. It essentially adopts an encoder-decoder structure with 3 `Conv-BatchNorm-ReLU` blocks and 2 `AvgPool` layers for down-sampling for the encoder and a symmetric structure for the decoder. Notably, to make the network aware of different sizes of synthetic datasets, we concatenate the size embedding to bottle-necked features after the encoder. Inspired by the positional embedding in Transformer models [47] and the time-step embedding in diffusion models [15, 39], we encode the size by sinusoidal signals and a learnable non-linear transformation function. Embedding features are replicated and expanded along the spatial axes before concatenation with features from the encoder.

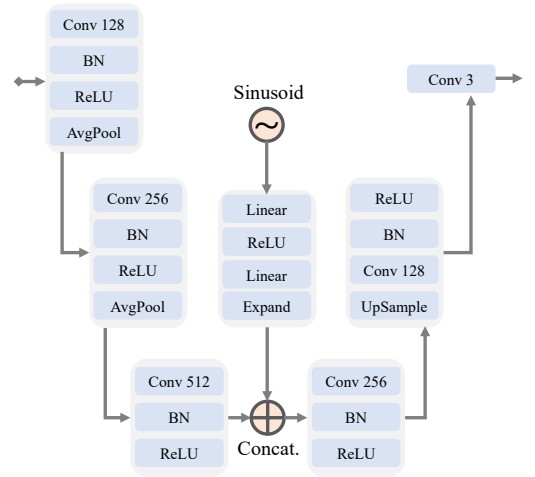

| Conv 128 |
| BN |
| ReLU |
| AvgPool |

Sinusoid ~

Conv 3

| Conv 256 |
| BN |
| ReLU |
| AvgPool |

| Linear |
| ReLU |
| Linear |
| Expand |

| ReLU |
| BN |
| Conv 128 |
| UpSample |

| Conv 512 |
| BN |
| ReLU |

⊕ Concat.

| Conv 256 |
| BN |
| ReLU |

Figure 7: Architecture of our generator network.

Table 7: List of hyper-parameters.

| Hyper-Parameter | Notation | Value |
|---|---|---|
| **Meta Learning Stage** | | |
| Number of Meta Testing Steps | $T'$ | 200,000 |
| Number of Meta Training Steps | $T$ | 5 |
| Maximal Number of Classes | $\max(C)$ | 100 |
| Minimal Number of Classes | $\min(C)$ | 10 |
| Maximal Number of Synthetic Samples | $\max(n_s)$ | 1,000 |
| Minimal Number of Synthetic Samples | $\min(n_s)$ | 10 |
| Number of Real Samples | $n_t$ | 2,000 |
| Learning Rate in Meta-Training | $\alpha$ | 1e-4 |
| Learning Rate in Meta-Testing | $\beta$ | 1e-5 |
| Parameter of Adam Optimizer | $(\beta_1, \beta_2)$ | $(0.9, 0.999)$ |
| Parameter of Cosine Learning Rate Scheduler | $\eta$ | 0.1 |
| **Adaptation Stage** | | |
| Number of Adaptation Steps | $T$ | 1,000 |
| Batch Size of Real Data | $n_t$ | 1,024 |
| Learning Rate of Generator | $\alpha$ | 1e-4 |
| Parameter of Adam Optimizer | $(\beta_1, \beta_2)$ | $(0.9, 0.999)$ |
| Parameter of Cosine Learning Rate Scheduler | $\eta$ | 0.1 |

| | Dataset | MNIST | | | FashionMNIST | | |
|---|---|---|---|---|---|---|---|
| | IPC | 1 | 10 | 50 | 1 | 10 | 50 |
| | Ratio (%) | 0.017 | 0.17 | 0.83 | 0.017 | 0.17 | 0.83 |
| Random | Acc. (%) | 64.9±3.5 | 95.1±0.9 | 97.9±0.2 | 51.4±3.8 | 73.8±0.7 | 82.5±0.7 |
| Full | Acc. (%) | | 99.6±0.0 | | | 93.5±0.1 | |
| DC [60] | Acc. (%) | 91.7±0.5 | 97.4±0.2 | 98.8±0.2 | 70.3±0.7 | 83.4±0.3 | 82.9±0.2 |
| | Time (sec.) | 157 | 3581 | 19811 | 155 | 3597 | 19829 |
| DSA [58] | Acc. (%) | 88.7±0.6 | 98.8±0.2 | 99.2±0.1 | 70.3±0.7 | 84.6±0.1 | 88.7±0.1 |
| | Time (sec.) | 172 | 3908 | 21259 | 173 | 3854 | 21118 |
| IDC [20] | Acc. (%) | 89.1±0.1 | 97.8±0.1 | 98.8±0.1 | 70.6±0.4 | 85.2±0.4 | 88.9±0.1 |
| | Time (sec.) | 22062 | 22798 | 28389 | 21929 | 23160 | 28499 |
| MTT [2] | Acc. (%) | 88.7±1.0 | 96.6±0.4 | 98.1±0.1 | 75.3±0.9 | 87.2±0.3 | 88.3±0.1 |
| | Time (sec.) | 3114 | 9323 | 9987 | 3107 | 9305 | 10092 |
| DM [59] | Acc. (%) | 89.7±0.6 | 97.5±0.1 | 98.6±0.1 | 71.5±0.5 | 83.8±0.2 | 88.2±0.3 |
| | Time (sec.) | 1115 | 1177 | 1457 | 1105 | 1172 | 1456 |
| FRePo [61] | Acc. (%) | 93.0±0.4 | 98.6±0.1 | 99.2±0.0 | 75.4±0.5 | 85.5±0.2 | 89.2±0.1 |
| | Time (sec.) | 6112 | 9174 | 21678 | 6115 | 8463 | 21549 |
| Ours | Acc. (%) | 91.3±0.2 | 97.8±0.2 | 99.0±0.0 | 73.8±0.8 | 84.7±0.2 | 88.3±0.1 |
| | Time (sec.) | **153**×40 | **392**×10 | **1012**×21 | **147**×42 | **432**×22 | **1005**×21 |

Table 8: Comparisons on test accuracy and running time with state of the arts on single-channel datasets. The acceleration marked by the red subscript is computed against the method with the best accuracy. IPC: Number of Images Per Class; Ratio: ratio of distilled images to the whole training set. Results demonstrate the cross-channel generalization ability of our meta generator.

# B  More Results

**Cross-Number-of-Channel Generalization:** In the meta learning stage, a meta generator is trained taking RGB images as input and output. Here, we demonstrate that it is also be feasible for the meta generator to be adapted for target datasets that have different numbers of channels. Specifically, we additionally train convolution layers for channel adaptation to map the number of channels from the original number to 3 and from 3 to original number at the beginning and the ending positions of the generator, respectively. The parameters of these adaptors are initialized from a uniform distribution and are optimized jointly with parameters of the generator.

Here, we conduct experiments on MNIST [25] and FashionMNIST [52] datasets. Both of them contain 10 classes with 60,000 single-channel images. Results are shown in Tab. 8 following the same comparison protocols as Tab. 1, where the generator in our method is adapted for 10,000 steps in each setting. Experiments demonstrate that our method can achieve comparable performance with those state-of-the-art ones in a significantly shorter period of time. The conclusion is the same as that in the main paper.

| IPC | 1 | 10 |
|---|---|---|
| Baseline | 28.20±0.77 | 48.26±1.26 |
| Ours | **36.51±0.47** | **49.20±0.10** |

| # of Classes | 20 | | 50 | |
|---|---|---|---|---|
| IPC | 1 | 10 | 1 | 10 |
| Baseline | 23.42±1.08 | 49.40±0.53 | 16.84±0.30 | 39.61±0.21 |
| Ours | 37.95±0.44 | 53.62±0.09 | 29.53±0.20 | 41.90±0.38 |

Table 9: Comparisons with the baseline FRePo on ImageNette under 128 resolution. Results demonstrate the cross-resolution generalization ability of our meta generator.

Table 10: Comparisons with the baseline FRePo on various CIFAR100 subsets. Results demonstrate the cross-number-of-classes generatlization ability of our meta generator.

| Dataset | CIFAR10 | | CIFAR100 | |
|---|---|---|---|---|
| IPC | 20 | 5 | 5 | 2 |
| DC | 41.8±0.6 | 25.9±0.4 | 13.3±0.3 | 6.7±0.2 |
| DSA | 41.5±0.4 | 27.6±0.2 | 14.9±0.3 | 8.1±0.1 |
| IDC | 51.9±0.5 | 30.2±0.4 | 13.3±0.3 | 11.0±0.1 |
| MTT | 55.9±0.3 | 29.8±0.4 | 26.7±0.5 | 13.7±0.3 |
| DM | 46.8±0.5 | 25.3±0.3 | 15.7±0.3 | 8.0±0.2 |
| FRePo | 59.1±0.7 | 38.3±0.9 | 30.0±0.6 | 19.9±0.3 |
| Ours | **60.8±0.4** | **46.1±0.8** | **30.6±0.3** | **25.4±0.5** |

Table 11: Comparisons with state of the arts on cross-IPC generalization.

| Dataset | IPC | DC [60] | DSA [58] | IDC [20] | MTT [2] | DM [59] | FRePo [61] | Ours |
|---|---|---|---|---|---|---|---|---|
| MNIST | 1 | **88.7±0.5** | 87.7±0.6 | 76.1±0.1 | 73.1±0.8 | 87.8±0.7 | 64.8±0.9 | 87.8±0.2 |
| | 10 | 96.2±0.2 | 96.7±0.1 | 95.1±0.1 | 92.8±0.2 | 96.2±0.1 | 96.3±0.1 | **97.2±0.1** |
| | 50 | 95.7±0.2 | 98.3±0.1 | 98.4±0.1 | 96.6±0.1 | 98.0±0.1 | 98.5±0.1 | **98.6±0.1** |
| FashionMNIST | 1 | 70.3±0.7 | 70.3±0.7 | 64.4±0.4 | 70.5±1.2 | 71.1±0.3 | 61.5±0.3 | **71.9±0.4** |
| | 10 | 79.8±0.2 | 79.0±0.3 | 82.9±0.2 | 80.1±0.5 | 83.0±0.1 | 81.2±0.2 | **83.4±0.2** |
| | 50 | 78.5±0.2 | 86.9±0.1 | 87.0±0.1 | 86.2±0.1 | 86.8±0.2 | 85.9±0.1 | **87.2±0.1** |
| CIFAR10 | 1 | 28.2±0.7 | 28.1±0.7 | 25.3±1.0 | 36.8±0.5 | 26.8±0.8 | 27.2±0.5 | **42.6±0.3** |
| | 10 | 39.7±0.5 | 48.7±0.3 | 49.5±0.3 | 50.8±0.5 | 48.8±0.2 | 49.4±0.3 | **58.9±0.4** |
| | 50 | 39.1±1.0 | 56.0±0.4 | 61.7±0.2 | 56.5±0.5 | 57.7±0.3 | 61.8±0.2 | **66.8±0.2** |
| CIFAR100 | 1 | 12.4±0.2 | 13.8±0.2 | 15.4±0.2 | 13.2±0.6 | 11.9±0.2 | 10.1±0.2 | **20.8±0.2** |
| | 10 | 21.1±0.2 | 31.3±0.4 | 28.9±0.3 | 30.2±0.4 | 30.0±0.4 | 26.6±0.4 | **32.2±0.3** |

Table 12: Comparisons with state of the arts on various benchmarks under the same number of training steps. IPC: Number of Images Per Class. Results demonstrate the superior efficiency of our method.

**Cross-Resolution Generalization:** Although the meta generator is trained under 32 resolution, it is possible for it to be adapted for datasets with different resolutions, thanks to the fully-convolutional architecture of the generator. We demonstrate the cross-resolution generalization performance on ImageNette [11], which contains 10 classes and 9,469 images. Following the FRePo baseline [61], we conduct experiments on 1 and 10 IPCs under 128 resolution. Results in Tab. 9 demonstrate the feasibility of such cross-resolution generalization.

**Cross-Number-of-Class Generalization:** Here, we conduct experiments on CIFAR100 subsets with random 20 and 50 classes respectively and compare the performance with the FRePo baseline [61]. Results in Tab. 10 demonstrate that the meta generator performs robustly on datasets with various numbers of classes.

**Cross-IPC Generalization:** For existing methods, when budgets for synthetic datasets change, they have to either repeat the time-consuming training loop of dataset distillation, which is inconvenient if not infeasible at all, or prune some synthetic data heuristically, which leads to inferior performance. For example, as shown in Tab. 11, on CIFAR10, if the original synthetic IPC is 50 and the new IPC becomes 20 or 5, random pruning would lead to unsatisfactory performance for existing methods. By contrast, the generator in our MGDD can work for arbitrary sizes of synthetic datasets once adapted, which makes it handle such scenarios better. We present another example on CIFAR100, the original IPC is 10 and the new IPC is 5 or 2.

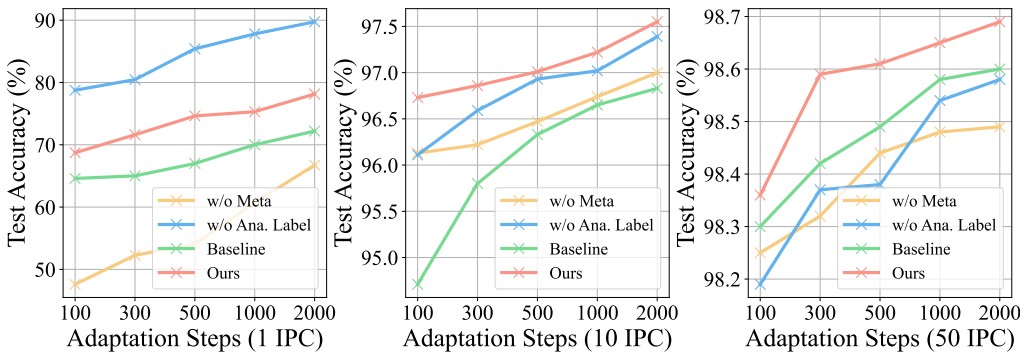

Figure 8: Performance of generators with various adaptation steps on MNIST.

**Comparisons under the Same Steps:** To better demonstrate the superiority of the proposed method, we compare our method with state of the arts with the number of training/adaptation steps controlled the same. As shown in Tab. 12, under 1000 steps, our method outperforms others significantly especially on relatively challenging datasets with more patterns, like CIFAR10 and CIFAR100. Furthermore, in Fig. 8, 9, 10, and 11, we visualize the performance of generators in each setting with different adaptation steps on MNIST, FashionMNIST, CIFAR10, and CIFAR100 datasets respectively as supplements to Fig. 4. It can be shown that our method can achieve the most satisfactory performance with only a limited number of adaptation steps compared with the baseline FRePo and generators from scratch, which indicates that the proposed method is more suitable for scenarios requiring high efficiency, like processing data streams. Note that for 1 IPC, we observe that using analytical labels would often lead to inferior performance compared with vanilla one-hot labels. We speculate that it is because soft labels by the analytical solution are relatively not good at leading the generator to synthesize class-discriminative patterns when the size of synthetic dataset is small. Thus, we do not use analytical labels for 1 IPC by default.

**Qualitative Results:** In Fig. 12, we supply qualitative visualization of initialized synthetic samples and results by generator under 1 and 10 IPC on CIFAR10 and 1 IPC on CIFAR100, as supplements to Fig. 6.

## C   Limitations and Future Works

Our MGDD method mainly focuses on the efficiency issue in existing methods. Although it can be demonstrated that our method can result in better performance in only limited time, it does not reduce the time and memory complexity of computing the matching metrics since we adopt the same objectives as previous approaches. When adapting for large synthetic datasets, it may still face the issue on GPU memory in existing works. Nevertheless, it is possible for our method to adapt on some small IPCs and then generalize to large synthetic datasets, as discussed in the main paper, which can serve as a remedy to this limitation. Besides, initialized samples of synthetic datasets come from real data, and results by generator still look somehow realistic, which may potentially make the method vulnerable to privacy attack, especially for data like personal information. Also, in scenarios like storing synthetic samples of human faces, the generator may break the integrity of faces and lead to an infringement of portrait rights if being misused.

Future works may focus on more effective training objective, training pipeline, and architecture of the generator in meta learning or/and adaptation stages to further improve the cross-dataset, cross-ipc, and cross-architecture generalization. It would also be valuable to extend the MGDD to other tasks and modalities beyond image classification and explore advanced input and output parameterizations of the generator.

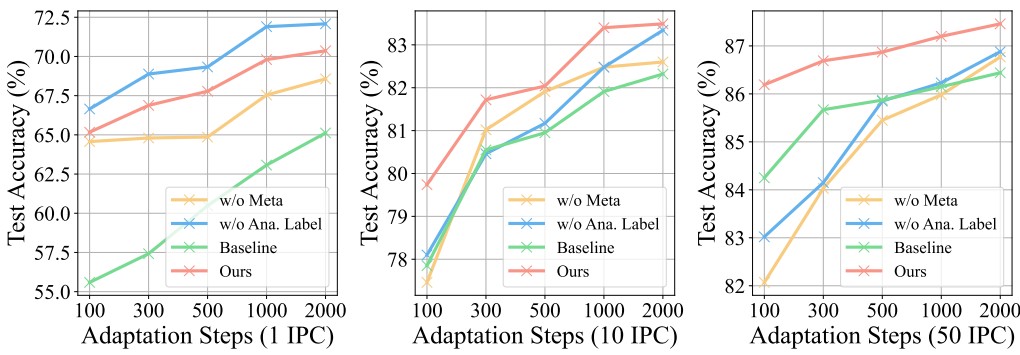

Figure 9: Performance of generators with various adaptation steps on FashionMNIST.

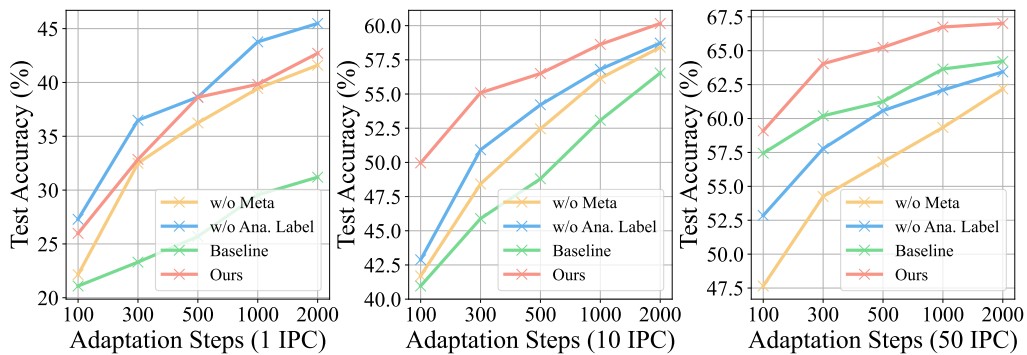

Figure 10: Performance of generators with various adaptation steps on CIFAR10.

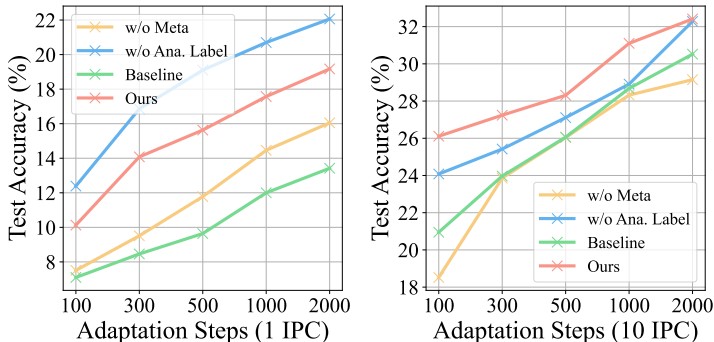

Figure 11: Performance of generators with various adaptation steps on CIFAR100.

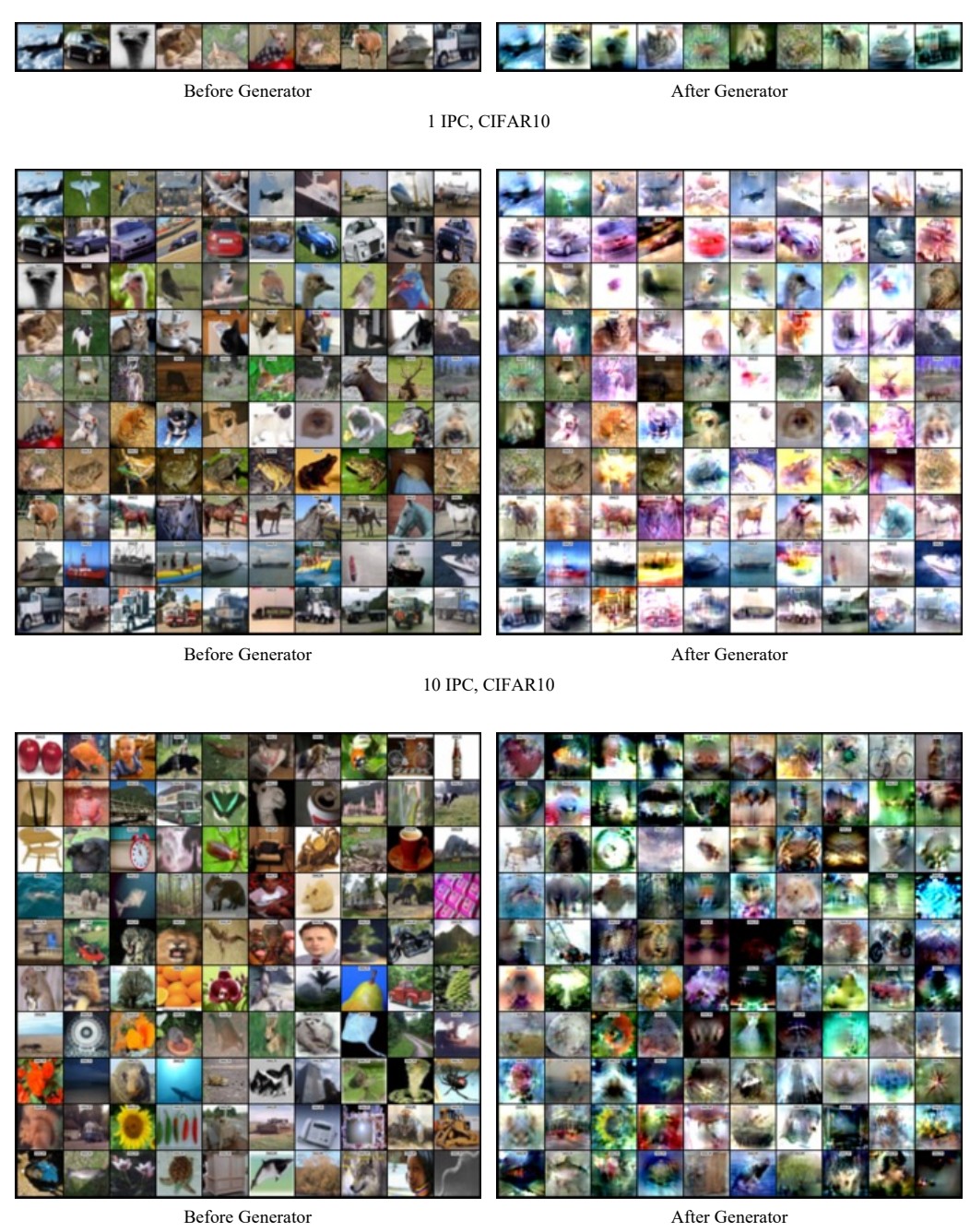

Figure 12: More visualizations of samples before and after generator on CIFAR10 and CIFAR100.

