# OpenReview forum: "MGDD: A Meta Generator for Fast Dataset Distillation"
_NeurIPS.cc/2023/Conference — NeurIPS 2023 spotlight_

### Official Review · Reviewer_Bfcg · 2023-07-05

**Soundness:** 4 excellent
**Presentation:** 4 excellent
**Contribution:** 4 excellent
**Rating:** 8
**Confidence:** 5

**Summary:**

In this paper, the authors address the efficiency concerns associated with dataset distillation. Initially, the authors identify the challenges arising from the dependencies within the intensive loop of forward-backward propagation across numerous neural networks, as well as the repetitive training procedures necessary when working with synthetic datasets of varying sizes. Motivated by these challenges, the authors propose an innovative approach to dataset distillation that eliminates the need for backpropagation. In this approach, synthetic labels are generated by analytically solving a least-squares problem in a feature space, while synthetic images are produced using a pre-learned meta generator. The authors provide theoretical evidence supporting the existence of a conditional generator capable of transferring synthetic samples from their original space to the desired space with an upper-bounded error. Experimental results demonstrate that the proposed backprop-free dataset distillation approach achieves comparable performance to state-of-the-art methods, while also providing a 22× acceleration. Additionally, the approach exhibits the flexibility of cross-size generalization.

**Strengths:**

I see the following positive traits:

+ This paper effectively tackles a crucial challenge in dataset distillation by addressing the cumbersome forward-backward propagation process required for effective distillation. The authors present an elegant solution rooted in meta-learning, which not only achieves an impressive speed improvement of 22 times but also maintains competitive performance levels.

+ The proposed method is supported by robust theoretical analysis, as evidenced by the proof of Theorem 1, where an upper-bounded error is derived, demonstrating the existence of a conditional generator capable of transferring the synthetic dataset from its original space to the desired feature space with an upper-bounded error.

+ The paper offers comprehensive algorithmic workflows, which are presented in both the main paper (Section 3.3, Algorithm 1) and the supplementary material (Section A: More Details, Algorithm 1) for handy reproducibility.

+ The paper is very well-written and easy to follow. The authors try to elaborate every detail while providing a clear overall picture.

+ The supplementary material of the paper offers an abundance of extensive details and additional experimental results including the visualization results, further validating the effectiveness of the proposed backprop-free dataset distillation method.


**Weaknesses:**

In general, this paper focuses on a significant problem, proposes an elegant solution, and the experimental results convincingly validate the effectiveness of the proposed approach. I only have some minor concerns as follows.

- The authors mention that the estimated acceleration factor includes the adaptation time. Given that the main emphasis of this paper is on time complexity, I would suggest the authors present the adaption time and feed-forward time separately in the experiments, to effectively illustrate the time cost involved in the process.

- I wonder how long it takes to train the meta generator on ImageNet1k.


**Questions:**

1. What is the physical adaption time for the results reported in the experiments?
2. How long does it require to train the meta generator on ImageNet1k?


**Limitations:**

The authors have thoroughly addressed the limitations in Section C of the supplementary material.

---

> ### Author Rebuttal · Authors · 2023-08-09
>
> We sincerely thank Reviewer Bfcg for the positive feedback on the manuscript and are very excited that the reviewer mentioned the strengths of effectiveness, theoretical analysis, algorithmic workflows, writing, and experiments. The questions are addressed below.
>
> * > What is the physical adaption time for the results reported in the experiments?
>
>   Thanks for the valuable question. The time reported includes the label-solving step, the adaptation procedure, and one forward propagation step. For the example of 1k steps on CIFAR10 with 10 IPC, the three terms take 7s, 56s, and 3ms respectively. The adaptation stage takes the major percentage, and the other two terms would not increase much with the size of synthetic datasets. The detailed analysis of the physical adaptation time reported in Tab. 1 of the main paper is as follows.
>
>   |      Setting      | Label Solving (s) | Adaptation (s) | Forward Propagation (ms) | Total Time (s) |
>   | :---------------: | :---------------: | :------------: | :----------------------: | :------------: |
>   |  CIFAR10 - 1 IPC  |         7         |      113       |           1.5            |      120       |
>   | CIFAR10 - 10 IPC  |         7         |      498       |           3.0            |      505       |
>   | CIFAR10 - 50 IPC  |         7         |      1388      |           11.0           |      1395      |
>   | CIFAR100 - 1 IPC  |         7         |      498       |           3.0            |      505       |
>   | CIFAR100 - 10 IPC |         7         |      2997      |           20.2           |      3004      |
>
>   We will include these details in the final version.
>
> * > How long does it require to train the meta generator on ImageNet1k?
>
>   Thanks for the insightful question. It requires 43 hours to train the meta-generator for 200,000 iterations on an A40 GPU. We will clarify this in the revision.
>
> Hope this will address the reviewer's questions, and thanks for the time, efforts, and positive review again.

---

### Official Review · Reviewer_CHTg · 2023-07-05

**Soundness:** 3 good
**Presentation:** 4 excellent
**Contribution:** 3 good
**Rating:** 7
**Confidence:** 4

**Summary:**

The reviewed paper makes a clear and novel contribution to the field of dataset distillation, which is expected to have a beneficial impact. The paper is well written, demonstrating a structured and easily understandable presentation. Notably, the empirical evaluation is exceptional, covering various aspects of the proposed method&rsquo;s effects, including evaluation of a potential application and examination of its combination with orthogonal methods.


**Strengths:**

The reviewed paper makes a clear and novel contribution to the field of dataset distillation, which is expected to have a beneficial impact. The paper is well written, demonstrating a structured and easily understandable presentation. Notably, the empirical evaluation is exceptional, covering various aspects of the proposed method&rsquo;s effects, including evaluation of a potential application and examination of its combination with orthogonal methods.

**Weaknesses:**

### Main Claim of being Backprop-Free

I disagree with the main claim of this work being backpropagation-free. The core mechanism of the *distillation* procedure in this work is the training of the meta-generator $g_{\omega}$ and its adaptation to a new target dataset. Both procedures rely on backpropagation. Yes, the actual generation of synthetic samples does not include backpropagation since it is a forward pass of a generative model with real samples as input. But that is like saying GANs are backprop-free &#x2013; sure, the generation of new samples does not require backpropagation, nevertheless the GAN needs to be trained, i.e. its parameters optimized, which relies on backpropagation for the generation of new samples to actually match the target data distribution. There is one, similar in concept, concurrent work to this paper: &ldquo;DiM: Distilling Dataset into Generative Model&rdquo; [1] which also distills the dataset into generative model to then later produce synthetic samples from this model &#x2013; similar to how in this work the meta-generator is trained to then produce synthetic samples. As explained in DiM and applicable to this work as well, the difference to standard dataset distillation methods is that the medium of information (into which the dataset is distilled) are not images but a parameterized model (from which one can then infer/sample images).

I am bit unsure of what to do now. I really like this work, it is novel, of high quality, easy to comprehend, and the evaluation is remarkable. But I disagree with the main claim of this being backprop-free, which is also part of the title. If this claim is removed (which I do not know if it is possible or if this would be too much of a change), I am happy to change my rating to accept. I am looking forward to discuss this issues with the authors and the other reviewers.


### Runtime Performance

In line 38-41 the authors state &ldquo;for FRePo, the method with the best performance requires above 3 hours to obtain a synthetic dataset with 1 image per class for CIFAR100&rdquo;. Where do these &ldquo;3 hours&rdquo; come from? Your Table 1 lists about 3 hours for FRePo on 1-IPC on CIFAR100 but this does not align with what is reported in the FRePo paper: &ldquo;When learning one image per class on CIFAR100, FRePo reaches a similar test accuracy (23.4%) to the second-best method (24.0%) in 38 seconds [&#x2026;]. Moreover, FRePo achieves 92% of its final test accuracy (26.4% out of 28.7%) in only 385 seconds.&rdquo;. I couldn&rsquo;t find the runtime for their &ldquo;final test accuracy&rdquo;, but even only taking their 26.4% result in 385 seconds (1-IPC CIFAR100) into consideration, FRePo beats the proposed method in terms of accuracy (24.66%) and is 1.3x faster than the proposed method (505s) &#x2013; opposed to x22 slower as claimed in the paper. Furthermore, the hardware used in FRePo (Quadro RTX 6000) is worse in terms of performance than the one used in this work (RTX 3090 for adaptation) which should only widen this discrepancy even more. Furthermore, the meta-training cost is not taken into account. Maybe I got something completely wrong here and I&rsquo;m happy to be corrected if I did, but since the x22 statement is repeated multiple times in this work I wanted to look into the details here.



### Minor Suggestions for Improvement

-   Figure 3-5: For improved visual consistency, you could try to make these figures the same height and width.
-   Paragraph of Line 296: I was shortly confused about which figure you were talking about &#x2013; another reference to Figure 3 would help here.

[1] Wang, K., Gu, J., Zhou, D., Zhu, Z.H., Jiang, W., & You, Y. (2023). DiM: Distilling Dataset into Generative Model. ArXiv, abs/2303.04707.


**Questions:**

-   What are the runtime costs of training the meta-generator?
-   Line 227: &ldquo;Moreover, we report the cost of time to derive synthetic datasets.&rdquo; -> Can you be more specific here? What includes &ldquo;derive synthetic datasets&rdquo;? Is this the adaptation procedure or does this inlcude the meta-training?
-   Line 200: &ldquo;For computation efficiency, the generator processes each sample independently&rdquo; -> Why should this be more efficient than batch-wise processing?
-   Line 202: &ldquo;The meta generator is trained by the Adam optimizer [15]&rdquo; -> In Algorithm 1 it is plain GD. Is this just out of simplicity in Algorithm 1 or am I missing something?
-   Line 238: &ldquo;On the contrary, the results of our method are produced by a generator in a one-stop fashion&rdquo; -> What do you mean by &ldquo;one-stop&rdquo; fashion? The meta-generator needs to be adapted to the target dataset which includes a certain number of forward-backward passes of $g_{\omega}$.


**Limitations:**

Limitations are addressed in the appendix. One main limitation which the work generally glances over is the meta-training procedure and its associated cost. All of the experimental evaluations have focused on the adaptation to a target dataset, but a core part of the proposed method lies in the meta-training of $g_{\omega}$.

---

> ### Author Rebuttal · Authors · 2023-08-09
>
> We are more than grateful to Reviewer CHTg for the positive feedback, valuable comments, and frank discussions regarding the manuscript. We are very delighted that the reviewer found our contribution novel, paper well written, and evaluation exceptional. For the concerns and questions, we would like to have the following discussions.
>
> * > Main claim of being backpropagation-free.
>
>   We thank the reviewer for pointing this out and fully understand the concern that both meta learning and fast adaptation stages require back propagation. Previously, we titled the method to be backpropagation-free since the meta learning is required only once for an infinite number of datasets and the adaptation is required only once for one dataset. Once adapted, the generator works in a feed-forward and backprop-free manner for different distillation requests, *e.g.*, different IPC and different initialization as shown in the experiment part, which is one of the most distinguishable features in this paradigm.
>
>   We do agree with the reviewer that the method is not purely backprop-free. Thus, to highlight the differences with prior works and the features in our method more clearly, we will change our title to "*Feed-Forward Dataset Distillation via Learning to Fast Adapt*", since 1) synthetic data in our approach are generated indeed in a feed-forward way although not purely backprop-free, and 2) the primary goal of this paper is exactly to find a meta generator that can be adapted to downstream datasets efficiently.
>
>   The reviewer mentioned a concurrent work DiM, which also uses generative models to produce synthetic data. However, the motivations, core methods, and functionalities are largely different. We will discuss more in the revision.
>
> * **Concerns in runtime performance.**
>
>   Thanks for pointing this out. The running time performance in Tab. 1 is from the time required for running the 500,000 iterations in the official code of FRePo [a] on our test environment, as mentioned in Line 237. Note that Tab. 1 is used for comparison with the final and official performance. In the following sub-sections, we would focus on detailed comparisons with it where **the number of iterations** are controlled as the same.
>
>   The reviewer mentioned a case that the baseline FRePo achieves better performance in less time.  The reason here is on the libraries used for implementation. Specifically, the official implementation of FRePo uses JAX, while our implementation uses PyTorch. There is a large gap in the efficiency of the two libraries. In general, for the baseline, the JAX version runs ~2 times faster than the PyTorch counterpart. In addition, as shown in Fig.2 of the rebuttal pdf for the mentioned CIFAR100 with 1 IPC, we find that the PyTorch counterpart of the baseline often results in inferior performance compared with the JAX version and the official JAX version is better to be adopted for detailed comparisons. To eliminate the effects of different libraries, we do not apply the running time as the metric and focus on the performance under the same number of iterations instead.
>
>   Moreover, we find that in many cases even though the running time is applied as the metric, where our method runs fewer steps, our method can still yield performance gain as shown in Tab. 1 of the rebuttal pdf. The results align with our primary goal: to enhance the performance of DD in a limited number of iterations.
>
> * > Figure 3-5: For improved visual consistency, you could try to make these figures the same height and width.
>
> * > Paragraph of Line 296: I was shortly confused about which figure you were talking about – another reference to Figure 3 would help here.
>
>   Thanks for the suggestions. Sure, we will fix them in the revision.
>
> * > What are the runtime costs of training the meta-generator?
>
>   Thanks for the valuable question. The time cost of training the meta-generator for 200,000 iterations on an A40 GPU is 43 hours. We will clarify this in the revision.
>
> * > Line 227: “Moreover, we report the cost of time to derive synthetic  datasets.” -> Can you be more specific here? What includes “derive synthetic datasets”? Is this the adaptation procedure or does this include the meta-training?
>
>   Thanks for the question. The time here includes the label-solving step, the adaptation procedure, and one feed-forward step. Meta-training is not included. We will be more specific in the final version.
>
> * > Line 200: “For computation efficiency, the generator processes each sample independently” -> Why should this be more efficient than batch-wise processing?
>
>   Thanks for the question. We indeed process samples batch-wisely. Here the "independently" means that the generator applies the same transformation function for all samples, and there is no interaction among different samples.
>
> * > Line 202: “The meta generator is trained by the Adam optimizer [15]” -> In Algorithm 1 it is plain GD. Is this just out of simplicity in Algorithm 1 or am I missing something?
>
>   Thanks for the question. This is just out of simplicity in Algorithm 1. We will clarify this in the revision.
>
> * > Line 238: “On the contrary, the results of our method are produced by a generator in a one-stop fashion” -> What do you mean by “one-stop” fashion? The meta-generator needs to be adapted to the target dataset which includes a certain number of forward-backward passes of $g_\omega$.
>
>   Thanks for the question. Here the "one-stop" means that once adapted the synthetic data can be generated in a single feed-forward time. We agree with the reviewer that it may result in some confusion. In the future, we will replace "a one-stop fashion" with "a limited number of adaptation steps" for better clarity.
>
> Thanks again for the insightful review. Hope our response can clear the reviewer's concerns and we are more than happy to have a discussion if there are more questions.
>
> [a] Dataset Distillation Using Neural Feature Regression, Zhou et al., NeurIPS 2022.

---

> > ### Comment · Reviewer_CHTg · 2023-08-11
> >
> > > The reviewer mentioned a concurrent work DiM, which also uses generative models to produce synthetic data. However, the motivations, core methods, and functionalities are largely different.
> >
> > Oh, yes, it is largely different -- I did not mean to compare DiM and the author's contributions but wanted to state that the way the DiM paper puts it is rather accurate: "Distillation into Images" vs. "Distillation into Models". The author's method thus falls into the "Distillation into Models" bucket.
> >
> >
> > > Once adapted, the generator works in a feed-forward and backprop-free manner for different distillation requests, e.g., different IPC and different initialization as shown in the experiment part, which is one of the most distinguishable features in this paradigm.
> >
> > I agree with the authors that this is a highly interesting and valuable contribution. Nevertheless, the *dataset distillation* happens during training and adaptation. The meta generator learns some form of distribution of the dataset **with backpropagation**. The *dataset generation* itself can then be seen as (conditionally) sampling from the learned distribution. Therefore, I think that even "Feed-Forward Dataset Distillation via Learning to Fast Adapt" is simply wrong as, again, the distillation itself happens during the meta generator training/adaption using backpropagation.
> >
> > The author's work could be viewed from a pretraining perspective: The meta generator is a model that is pretrained, which can then be efficiently adapted to target distributions. Therefore, the authors could think of a different name along the lines of "Pretraining Dataset Distillation".
> >
> > > Rebuttal PDF
> >
> > I want to thank the authors for providing additional empirical results in the rebuttal PDF. The extensive additional dataset evaluation makes the results of the authors contributions even stronger, and the time vs. accuracy plots alleviated my concerns on the performance comparison.
> >
> > Having read the other reviews and rebuttals and in the hope that the authors will find a better suiting name for their work and get rid of the "backprop-free" claim, I will update my rating from 4 to 7.
> >
> > (If anyone is wondering about the "drastic" increase: I was already at a 6 internally but remained with reject due to my concerns on the backprop-free claims. Since the authors have shown to be open to the discussion, my correct score should be 6 and with the reviews and rebuttals this score moved to a 7.)

---

> > > ### Author Response · Authors · 2023-08-11
> > > **Thanks for the Reply and an Update on the Title**
> > >
> > > We greatly appreciate the further discussions and constructive comments from Reviewer CHTg. We are encouraged and excited to see that most concerns have been alleviated. Specifically, we would like to thank the reviewer for pointing out "pretraining", which is indeed an insightful perspective to understand our contribution. As advised, to better reflect the "pretraining" nature of our work, we will change our title to *MGDD: A Meta Generator for Fast Dataset Distillation*, given that the pretraining is instantiated with a meta-learning algorithm.
> > >
> > > Again, we thank Reviewer CHTg for the valuable comments and suggestions, which have contributed significantly to the enhancement of our manuscript.

---

### Official Review · Reviewer_pTHr · 2023-07-09

**Soundness:** 3 good
**Presentation:** 2 fair
**Contribution:** 3 good
**Rating:** 6
**Confidence:** 4

**Summary:**

- This paper proposing a meta-learning version of dataset distillation (DD)
- The main idea is that dataset distillation is slow, and meta-learning can be used to amortize the dataset distillation procedure
- The goal a function $g_\omega (X_{init}) = X_S$, which changes an initialized distilled dataset to the final one
- The labels are learned to minimize the KRR loss, using the feature space of randomly initialized networks
- Compared to MAML, in this case $\omega_{init}$ is functionally equivalent to $\theta_{init}$ in MAML, and $\omega$ is updated for a few parameter steps. In this sense the goal is to learn an optimal $\omega_{init}$

**Strengths:**

- A meta-learning approach to DD is new, and is valuable considering that current dataset distillation algorithms are very slow (and often the cost of doing DD is much much greater than the cost of training networks on the full data, often negating their value)

- The improvements in computation time are very dramatic, and the cost in terms of performance on the reported baselines is low

- I am very impressed with the cross-IPC generalization (line 281). Being able to growing/shrink a distilled dataset in response to changing needs without completely retraining a network is very desirable, and is lacking with current DD algorithms

**Weaknesses:**

- Poor presentation: I personally found the presentation very confusing. In particular, section 3.2 seemed to add more confusion to the paper. I think and outline of the method (3.3) should have been presented before this theory section. In particular I found the idea of the parameterized transfer function $g_\omega$ to have been irrelevant until after reading section 3.3. In regards to presentation, I think it needs to be more clearly stated that $\omega$ is adapted in training, and that you are learning the initialization of $\omega$, because section 2.3 gives the impression that $\omega$ is independent of $X_t$ (line 136 - "without any dependence on $X_t$"), which is misleading since it updated using $X_t$. I think discussing this in comparison to MAML would be useful since readers are probably familiar with MAML.

- Limited evaluation in terms of datasets. As a major selling point of a meta-approach to dataset distillation is being able to quickly adapt to unseen datasets, it seems necessary to test more than just CIFAR-10 and CIFAR-100. I understand that this is difficult as most dataset distillation algorithms don't benchmark on a lot of datasets (for example apart from CF-10/100, MNIST/F-MNIST, CUB-200 and tinyimagenet/imagenet subsets seem to be the only other common ones), and since this algorithm meta-trains on imagenet, some of these are eliminated. However, I think it is particularly important for this paper to include more datasets. Even mixed datasets (i.e. images from collections of different datasets) may be sufficient

- More discussion of meta-training costs. In particular, how long does meta-training take? It also seems that $\omega$ is optimized for several hundred iterations, and this sounds like in order to store this is memory would be quite costly. I see that in lines 200-205, it is mentioned that a A40 (a singificantly stronger GPU) is used for meta-training. This cost should be stated more clearly in the limitations

-Minor:
 - I think there are typos in eq (2) (line 125). I'm not sure $\theta^*$ is meant to appear here.
- Missing citation. Lines 129-130: the method of solving for the optimal labels was first proposed as "label solve" in [1]

[1] Timothy Nguyen, Zhourong Chen, & Jaehoon Lee (2021). Dataset Meta-Learning from Kernel Ridge-Regression. In International Conference on Learning Representations.


**Questions:**

- I'm not sure if Theorem 1 is particular relevant to the method. As mentioned in the weaknesses, this theorem takes $\omega$ to be independent of $X_t$ but in reality it is not because of meta-adaptation. Furthermore, there is not really interpretation of what the upper bound means. What does the right hand side term in eq. 6 mean? In particular why does it make sense to consider the product $f_{\theta^*}(X_s)W^{\theta}_s$, considering that one is based on the feature space $\theta$ and the other based on $\theta^*$? While the feature spaces are expected to induce similar kernels, multiplying the features together directly does not make sense because they could be randomly permuted versions of one another, for example

- What is the effect of using a larger learning rate for FRePo in figure 4?

- Why do you base $Y_s^*$ on the initialization images $X_{s}$ instead of the transformed images $g_\omega(X_s)$ in Alg. 1 line 5?

**Limitations:**

See weaknesses

---

> ### Author Rebuttal · Authors · 2023-08-09
>
> We would like to express our sincere gratitude for the pertinent comments and suggestions of Reviewer pHTr. We are glad that the reviewer finds our method novel and valuable, improvements in efficiency dramatic, and cross-IPC generalization impressive. The concerns, which withhold the reviewer's rating, are addressed as follows.
>
> * **Presentation.**
>
>   We thank the reviewer's feedback from a reader's perspective. Previously we organized Sec. 3 as this form because Sec. 3.2 and Sec. 3.3 correspond to the two steps in our method **in sequence**: we first get the synthetic label and then use the generator to generate synthetic samples. We agree with the reviewer that it may result in some confusion, especially for the theoretical analysis part. Thus, in the next version, we plan to adjust the organization as follows. Sec. 3.2 and Sec. 3.3 will still introduce the technical methods of the two steps respectively. Theorem 1, which appears in Sec. 3.2 at present, will be moved to a new sub-section 3.4. All the details mentioned by the reviewer will be clarified. We hope that the new organization can improve the presentation of our method.
>
> * **More datasets.**
>
>   We thank the reviewer for the pertinent suggestion and agree that involving more datasets could better demonstrate the effectiveness of our method. Thus, we further add evaluations on PACS, PathMNIST, BloodMNIST, and CUB200 datasets here. For some datasets, like Art painting, Cartoon, and Sketch in PACS, and PathMNIST and BloodMNIST for medical image classification, there are large domain gaps with the pre-training dataset ImageNet. We also evaluate the method on mixed datasets as the reviewer suggested: *P+A+C+S* indicates that we mix the 7 classes from the 4 domains in PACS and form a 28-class dataset, and *PathBloodMNIST* indicates that we mix the 9 classes in PathMNIST and 8 classes in BloodMNIST and form a 17-class dataset. The results of our method, our method without the meta generator, are shown in Tab. 2 of the rebuttal pdf. The results suggest that the meta generator can be generalized to unseen datasets successfully and is robust to domain variations.
>
> * **More discussion of meta-training costs.**
>
>   Thanks for pointing this out. Some details of the meta training are shown in Tab. 1 of the supplement. The time and GPU memory costs of meta training are 43 hours and 15.8 GB respectively. The memory would not increase with more meta-training steps since we use the first-order MAML (FOMAML) [b]. We will clarify these details clearly in the final version.
>
> * > I think there are typos in eq (2) (line 125). I'm not sure $\theta^*$ is meant to appear here.
>
>   Thanks for the detailed inspection and $\theta^*$ should be $\theta$ here. We will correct this in the revision.
>
> * > Missing citation. Lines 129-130: the method of solving for the optimal labels was first proposed as "label solve" in [1].
>
>   We thank the reviewer for bringing this to our attention and acknowledge that [1] first proposed a simple yet effective label solving method. We will definitely add proper discussions with [1] mentioned by the reviewer in the revision, and would like to mention here that there is a small difference in equations: $X_sX_s^\top(X_tX_s^\top)^+Y_t$ in [1] v.s. $X_sX_t^+Y_t$ in ours. We empirically find that the latter works slightly better in practice, *e.g.*, 57.8% v.s. 58.9% accuracy on CIFAR10 with 10 IPC and 1k steps.
>
> * **Questions about Theorem 1.**
>
>   Thanks for the insightful questions. On the one hand, Theorem 1 is used to intuitively demonstrate the primary feasibility of our two-step pipeline: solving labels and transferring samples. Even if $\omega$ is not dependent on $X_t$, we can obtain an upper-bounded error. In practice, if we further have the information of $X_t$, the error is expected to be lower. Please refer to the discussion between Lines 146 to 152.
>
>   On the other hand, for the sake of better clarity, we would like to modify the discussion in Eq. 6 in the revision and use the result in Eq. 5 as the error bound. In fact, Eq. 5 is valid for any transfer function $g_\omega$. Since 1) $\theta^*$ is unknown in practice, and 2) it is challenging to solve the optimal $g_\omega$ by minimizing the first term of the right-hand side of Eq. 5 even if $\theta^*$ is known, in this paper we approach $g_\omega$ through adaptation from a meta model encoding some meta knowledge.
>
> * > What is the effect of using a larger learning rate for FRePo in figure 4?
>
>   Thanks for the question. In Fig. 1 of the rebuttal pdf, we try larger learning rates for FRePo. Using larger learning rates for FRePo benefits the initial convergence. However, the performance may get saturated too early, and it would not increase, or even decrease, in later adaptation stages. Our method still demonstrates the advantages in terms of overall performance.
>
> * > Why do you base $Y_s^*$ on the initialization images $X_s$ instead of the transformed images $g_{\omega}(X_s)$ in Alg. 1 line 5?
>
>   Thanks for pointing out an alternative solution that first obtains $g_\omega(X_s)$ and then solves the label in some space parameterized by $\theta$. In fact, label solving can find the optimal label for some given $\theta$. However, our final goal is to minimize the error on $\theta^*$, which is not known and is approximated through iterative adaptation. Thus, it is better to place the adaptation stage in the last step before the label solving, which aligns with our final goal $\theta^*$ rather than $\theta$. The experiment results indeed verify this, *e.g.*, 54.0% v.s. 58.9% accuracy on CIFAR10 with 10 IPC and 1k steps.
>
> We would like to thank Reviewer pTHr again for the in-depth reviews. We would definitely love to further interact with the reviewer if there are any further questions.
>
> [a] Dataset Distillation Using Neural Feature Regression, Zhou et al., NeurIPS 2022.
>
> [b] Model-Agnostic Meta-Learning for Fast Adaptation of Deep Networks, Finn et al., ICML 2017.

---

> > ### Comment · Reviewer_pTHr · 2023-08-12
> >
> > Thanks for running the new experiments. I have a few followup questions/comments.
> > - The results on the new dataset are very interesting. It would be useful to have FrePo trained until convergence on the new datasets (as it seems that you only have FRePo stopped after a few iterations). Additionally, are the FrePo results with the optimal learning rate or with the off-the-shelf one? As we see in fig 1 in the rebuttal pdf, it seems that the FrePo learning rate makes a large difference.
> >
> > - Regarding label solve - thanks for pointing out the difference between label solve and your algorithm - I did not notice it at first. I am rather surprised that minimizing the upper bound in eq. 2 actually works better than minimizing the far left-hand-side, as label solve does.
> >
> > - Meta training costs - I think mentioning that you are using FOMAML is quite important because it greatly affects the memory costs of the algorithm
> >
> > I would inclined to increase my score if the authors include results on the new datasets with FrePo (or any other baseline) trained until convergence, as most of my questions have been addressed.

---

> > > ### Author Response · Authors · 2023-08-15
> > > **Thanks for the Valuable Followup Comments + New Experimental Results**
> > >
> > > We truly appreciate Reviewer pTHr for engaging with us and are very glad to see that most of the concerns have been addressed. Here, we would like to address the remaining ones as follows.
> > >
> > > * > Are the FrePo results with the optimal learning rate or with the off-the-shelf one?
> > >
> > >   Thanks for the insightful question. Indeed, we have noted that the performance is sensitive to the learning rate. On the one hand, the default learning rate (0.0003) is somewhat small to get a satisfactory performance in a limited time. On the other hand, with larger learning rates like 0.003, the performance may get saturated early and decrease in later adaptation stages. Therefore, we empirically set the learning rate as 0.001 in these new experiments.
> > >
> > > * > Results on the new datasets with FrePo (or any other baseline) trained until convergence.
> > >
> > >   Thanks for the valuable question concerning the final performance. Our major focus in this paper is on the dataset distillation performance in a limited number of steps. Thus, we mainly report the few-step performance of the baseline and our method in the rebuttal pdf. Nevertheless, we would also be happy to supplement the performance upon convergence here as mentioned by the reviewer. The datasets used are consistent with those in the rebuttal pdf. The results of the DSA [a], DM [b], and FRePo [c] baselines and ours on PACS are as follows.
> > >
> > >   |       | IPC  |         P          |         A          |         C          |         S          |      P+A+C+S       |
> > >   | :---: | :--: | :----------------: | :----------------: | :----------------: | :----------------: | :----------------: |
> > >   |  DSA  |  1   |   54.67$\pm$1.53   |   31.22$\pm$0.63   |   55.05$\pm$1.46   |   41.33$\pm$0.82   |   41.02$\pm$0.15   |
> > >   |  DM   |  1   |   53.47$\pm$0.77   |   28.29$\pm$1.14   |   48.59$\pm$0.66   |   36.26$\pm$1.16   |   32.81$\pm$0.33   |
> > >   | FRePo |  1   | **68.64$\pm$0.52** |   41.59$\pm$1.53   |   59.38$\pm$0.93   |   52.85$\pm$0.51   |   44.87$\pm$0.28   |
> > >   | Ours  |  1   |   66.77$\pm$1.81   | **44.24$\pm$0.36** | **64.48$\pm$0.68** | **53.40$\pm$1.58** | **48.39$\pm$0.49** |
> > >   |  DSA  |  10  |   78.64$\pm$0.37   |   56.30$\pm$0.35   |   72.06$\pm$0.15   |   70.31$\pm$0.55   |   61.05$\pm$0.32   |
> > >   |  DM   |  10  |   71.04$\pm$0.17   |   47.22$\pm$1.03   |   66.70$\pm$0.76   |   67.96$\pm$0.30   |   56.51$\pm$0.26   |
> > >   | FRePo |  10  | **85.40$\pm$0.30** | **60.64$\pm$1.64** |   78.10$\pm$0.54   |   76.87$\pm$1.09   |   65.38$\pm$0.48   |
> > >   | Ours  |  10  |   82.27$\pm$0.89   |   59.47$\pm$1.20   | **78.88$\pm$0.15** | **76.91$\pm$0.25** | **65.76$\pm$0.26** |
> > >
> > >   Results on CUB-200 and the medical datasets are as follows, where $^*$ indicates results from the FRePo paper [c].
> > >
> > >   |       | IPC  |        CUB-200         |     PathMNIST      |     BloodMNIST     |   PathBloodMNIST   |
> > >   | :---: | :--: | :--------------------: | :----------------: | :----------------: | :----------------: |
> > >   |  DSA  |  1   |   1.29$\pm$0.09$^*$    |   22.09$\pm$1.65   |   55.49$\pm$1.27   |   33.86$\pm$0.48   |
> > >   |  DM   |  1   |   1.61$\pm$0.06$^*$    |   41.87$\pm$1.12   |   62.02$\pm$2.92   |   43.20$\pm$0.99   |
> > >   | FRePo |  1   |   12.41$\pm$0.20$^*$   |   64.74$\pm$1.10   |   71.17$\pm$0.72   |   64.57$\pm$1.63   |
> > >   | Ours  |  1   |   **12.51$\pm$0.27**   | **70.25$\pm$0.11** | **72.93$\pm$1.70** | **65.30$\pm$0.44** |
> > >   |  DSA  |  10  |   4.54$\pm$0.26$^*$    |   51.93$\pm$1.27   |   80.28$\pm$0.46   |   68.31$\pm$0.54   |
> > >   |  DM   |  10  |   4.38$\pm$0.16$^*$    |   67.86$\pm$1.05   |   79.68$\pm$0.71   |   72.83$\pm$0.84   |
> > >   | FRePo |  10  | **16.84$\pm$0.12$^*$** |   77.44$\pm$1.53   | **85.81$\pm$0.56** |   76.58$\pm$0.25   |
> > >   | Ours  |  10  |     15.05$\pm$0.37     | **78.21$\pm$1.42** |   84.47$\pm$0.23   | **77.07$\pm$0.98** |
> > >
> > >   Through these results, we can observe that our method consistently outperforms the DSA and DM baselines, and generally outperforms the FRePo baseline when the distilled datasets are small, *e.g.*, 1 image per class. For larger distilled datasets, the number of parameters grows proportionally to the number of synthetic images; by contrast, our method optimizes a generator model and the number of learnable parameters does not increase with the size of synthetic datasets. As such, the performance gain from the increased IPC is not as significant as the baseline. Nevertheless, our method still yields comparable performance in these cases.
> > >
> > >   We sincerely hope the new results can address the reviewer's questions and thank the reviewer again for the constructive feedback.
> > >
> > > [a] Dataset Condensation with Differentiable Siamese Augmentation, Zhao et al., ICML 2021.
> > >
> > > [b] Dataset Condensation With Distribution Matching, Zhao et al., WACV 2023.
> > >
> > > [c] Dataset Distillation Using Neural Feature Regression, Zhou et al., NeurIPS 2022.

---

> > > > ### Comment · Reviewer_pTHr · 2023-08-15
> > > >
> > > > Thanks for the new results. Since the new results address the majority of my concerns, I will be increasing my score to a 6. Like reviewer CHTg, I also would agree that a title change would better reflect the contributions of the paper.

---

### Official Review · Reviewer_jcg2 · 2023-07-11

**Soundness:** 3 good
**Presentation:** 3 good
**Contribution:** 3 good
**Rating:** 6
**Confidence:** 4

**Summary:**

This paper proposes a meta-learning conditional generator method that takes some initialization of synthetic images and quickly learns to render it for Dataset Distillation purposes. The authors extensively verified the performance and show that the proposed method is substantially faster and achieves good performance.

**Strengths:**

+ This paper is well motivated and a good contribution to the Dataset Distillation community
+ The proposed algorithm is interesting. The algorithm is almost like treating the conditional generator as an updater/optimizer that learns to iteratively refine the synthetic set for a customized loss
+ The experimental results support the claims

**Weaknesses:**

- From figure 6, it looks like the updated images aren't moving a lot after the refinement by generator. Could the authors explain the potential reasons?
- The performance of DD are eventually not better than previous results.
- The method requires an additional neural network component for update/refinement

Misc. line 40 RTP -> RTP[7], missing citation number.

**Questions:**

See above.

**Limitations:**

Didn't find the equivalent paragraph for limitation. The authors should add and highlight it?

---

> ### Author Rebuttal · Authors · 2023-08-09
>
> We sincerely appreciate Reviewer jcg2 for the constructive comments. We are happy that the reviewer finds our work well-motivated, the method interesting, and the experiments extensive. We would like to address the concerns and questions reflected in the review below.
>
> * > Why do the updated images not move a lot after the refinement by the generator?
>
>   Thanks for the insightful question. The reason is on the training objective of DD we apply, *i.e.*, the NFR loss proposed in FRePo [a], where the visualizations of final distilled images are somewhat realistic and do not move much from the initialization. Thus, we assume that the image-to-image transformation is not complex and can be learned with a simple auto-encoder structure. Since operations in the generation model, like Conv and BN, are all global-wise, *i.e.*, different spatial locations in a feature map share a common transformation function, it is understandable that basic semantic structures do not change too much by the generator.
>
> * > The performance of DD is eventually not better than previous results.
>
>   Thanks for the comment. On the one hand, the major focus of the work is on the efficiency, or the performance in a limited number of iterations, which has been extensively verified in the main paper and the supplement. On the other hand, if enough time is given, the results can surpass previous ones in some cases. Here are some examples on CIFAR10:
>
>   |        IPC        |       1        |       10       |       20       |
>   | :---------------: | :------------: | :------------: | :------------: |
>   | Acc. of FRePo (%) | 43.24$\pm$0.32 | 65.76$\pm$0.72 | 67.60$\pm$0.49 |
>   | Acc. of Ours (%)  | 48.01$\pm$0.43 | 66.01$\pm$0.15 | 68.07$\pm$0.10 |
>
> * > The method requires an additional neural network component for update/refinement.
>
>   Thanks for pointing this out. Indeed, the proposed method comprises an additional generator, which introduces additional overhead for computing the matching metrics and it does not reduce the complexity. As mentioned in Sec. C of the supplement, we would like to view this as a limitation. Nevertheless, the introduced memory consumption is acceptable in fact, *e.g.*, 3182 MB v.s. 3288 MB for CIFAR10 with 1 IPC. Moreover, as discussed in Lines 90 to 92 of the supplement, it is possible for our method to adapt on some small synthetic datasets and then generalize to larger ones, which can serve as a remedy to this limitation.
>
> * > Misc. line 40 RTP -> RTP[7], missing citation number.
>
>   Thanks for the detailed review. We will definitely fix this in the next version.
>
> * > Didn't find the equivalent paragraph for limitation.
>
>   Thanks for the suggestion. Please refer to Sec. C of the supplement for details and we will highlight it in the final version.
>
> We would like to thank Reviewer jcg2 again for the valuable feedback. Hope our responses address the reviewer's concerns and we are happy to answer additional questions if there are.
>
> [a] Dataset Distillation Using Neural Feature Regression, Zhou et al., NeurIPS 2022.

---

### Author Rebuttal · Authors · 2023-08-09

We would like to express our heartfelt gratitude to all reviewers for their constructive reviews. The questions and concerns are addressed in the rebuttal under the Official Review block of each reviewer. We would like to upload a pdf file here, denoted as "rebuttal pdf" in the following parts. The file contains the following tables and figures that cannot fit into the rebuttal blocks:

* Fig. 1: Comparisons with the baseline FRePo using different learning rates. (Reviewer pTHr)
* Fig. 2: Difference between JAX and PyTorch implementations of the baseline FRePo. (Reviewer CHTg)
* Tab. 1: Comparisons with the baseline FRePo using running time as the metric. (Reviewer CHTg)
* Tab. 2: Evaluations on more benchmarks. (Reviewer pTHr)

We are more than happy to discuss with the reviewers if there are more questions. And thanks for the time and effort in reviewing this article again!

Best Regards,
Authors of Submission 5629

---

### Decision · Program_Chairs · 2023-09-21

**Decision:**

Accept (spotlight)

**Comment:**

This paper proposes a novel dataset distillation scheme that involves a meta-generator capable of generating synthetic datasets for a new dataset by adapting to it with only a few gradient update steps. Specifically, the generator is conditioned on the initially distilled dataset and generates synthetic data from it, while obtaining labels for the generated data points by solving a least-square problem in a feature space. The authors provide theoretical analysis demonstrating the existence of a conditional generator capable of transferring the synthetic dataset from its original space to a desired feature space with an upper-bounded error. The experimental results show that the proposed method can achieve significant improvements in computation time and generalization performance.

All reviewers unanimously express positivity about accepting the paper. They find the idea of adapting a meta-generator to each dataset novel and effective, the theoretical analysis sound, and the experimental results strong. The majority of the reviewers also find the paper well-written and mostly clear, while a few have concerns regarding the organization and clarity in certain sections. The reviewers, as well as the area chair, unanimously agree that the proposed scheme is novel and solves the practical challenge of high computation cost when applying dataset distillation to real-world scenarios. Therefore, the proposed work makes a clear and novel contribution.

However, there were some concerns regarding the misleading title (i.e., "backprop-free" while the generator should still be trained with backpropagation), the lack of discussion on the meta-training cost, and some unclear parts in the report of the runtime performance. Yet, the authors' rebuttal has addressed most of the concerns. The authors are strongly advised to incorporate the discussions with the reviewers into the final revision of the paper